# How does Labeling Error Impact Contrastive Learning? A Perspective from Data Dimensionality Reduction

Jun Chen [1]   Hong Chen [1 2]   Yonghua Yu [3]   Yiming Ying [4]

## Abstract

In recent years, contrastive learning has achieved state-of-the-art performance in the territory of self-supervised representation learning. Many previous works have attempted to provide the theoretical understanding underlying the success of contrastive learning. Almost all of them rely on a default assumption, i.e., the label consistency assumption, which may not hold in practice (the probability of failure is called labeling error) due to the strength and randomness of common augmentation strategies, such as random resized crop (RRC). This paper investigates the theoretical impact of labeling error on the downstream classification performance of contrastive learning. We first reveal several significant negative impacts of labeling error on downstream classification risk. To mitigate these impacts, data dimensionality reduction method (e.g., singular value decomposition, SVD) is applied on original data to reduce false positive samples, and establish both theoretical and empirical evaluations. Moreover, it is also found that SVD acts as a double-edged sword, which may lead to the deterioration of downstream classification accuracy due to the reduced connectivity of the augmentation graph. Based on the above observations, we give the augmentation suggestion that we should use some moderate embedding dimension (such as 512, 1024 in our experiments), data inflation, weak augmentation, and SVD to ensure large graph connectivity and small labeling error to improve model performance.

[1]College of Informatics, Huazhong Agricultural University, Wuhan, China [2]Engineering Research Center of Intelligent Technology for Agriculture, Ministry of Education, China [3]College of Engineering, Huazhong Agricultural University, Wuhan, China [4]School of Mathematics and Statistics, University of Sydney, NSW, Australia. Correspondence to: Hong Chen <chenh@mail.hzau.edu.cn>.

*Proceedings of the 42nd International Conference on Machine Learning*, Vancouver, Canada. PMLR 267, 2025. Copyright 2025 by the author(s).

## 1. Introduction

Contrastive learning, as an emerging self-supervised learning paradigm, has achieved remarkable performance by leveraging data without label information (He et al., 2020; Chen et al., 2020c; Jang & Wang, 2023; Wang et al., 2023b; Ji et al., 2023). Typically, this learning framework entails formulating an auxiliary contrastive task endowed with pseudo-labels, which aims to maximize the similarity between two samples augmented from the same original sample while minimizing the similarity between samples augmented from different original samples (Chen et al., 2020b).

Recently, some studies have delved into the theoretical mechanism underlying the empirical success of contrastive learning (Mikolov et al., 2013; Pagliardini et al., 2018; Jean et al., 2019; Arora et al., 2019; Jing et al., 2022; Wang et al., 2022; Lei et al., 2023). Generally speaking, they acquiesce that the labels of two augmented samples generated from the same original sample remain consistent, which is referred to as the label consistency assumption (Wang et al., 2022). Particularly, Wang et al. (2022) stated that label consistency is a natural and minimal assumption that is likely to hold in practice, and established the upper and lower bounds of downstream classification risk for contrastive learning only requiring intra-class samples have similar augmented views (called intra-class augmentation overlap, see Figure 1 (b)). Nevertheless, given that data augmentation process is random, some strong augmentation strategies like RRC may lead to the lose of semantic-related information (Zang et al., 2024), which undermines label consistency and gives rise to labeling error (Tamkin et al., 2023). For instance, Figure 1 (a) shows that several images augmented from a dog image using RRC possess different labels (dog, ship, fog and blanket). In such a situation, it can be observed that inter-class samples also exhibit the augmentation overlap phenomenon (see Figure 1 (c)). This inconsistent label phenomenon (called labeling error) motivates us to develop new theoretical analysis for contrastive learning to better understand the interplay among data augmentation, labeling error, and downstream classification performance.

Recently, the analysis framework of HaoChen et al. (2021) pioneered the consideration of the labeling error caused by data augmentation, which revealed the first dependencies

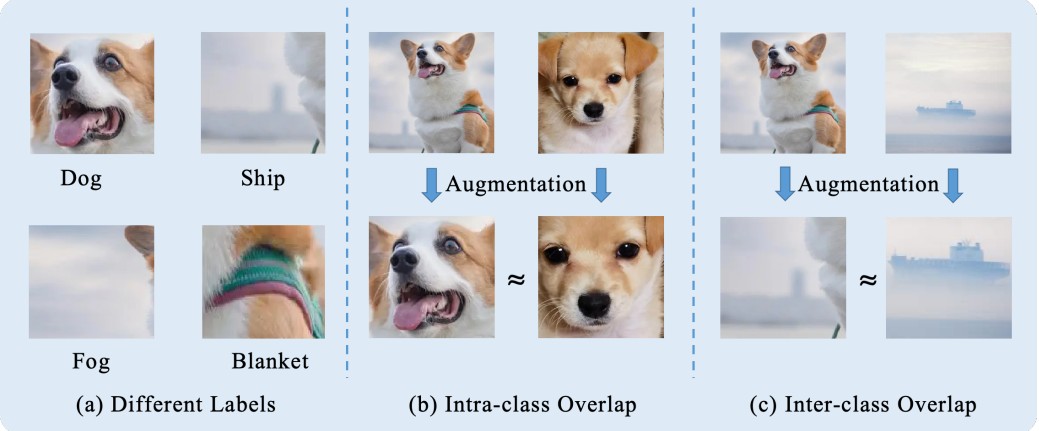

Dog        Ship

Fog        Blanket

Augmentation        Augmentation

≈        ≈

(a) Different Labels        (b) Intra-class Overlap        (c) Inter-class Overlap

*Figure 1.* (a) Four images augmented from a single dog image using RRC have different labels (dog, ship, fog and blanket). (b) Augmented views of different intra-class samples may overlap. For instance, two views labeled as "dog" might be augmented from different dog images. (c) Augmented views of different inter-class samples may also overlap. For instance, two views labeled as "ship" might be augmented from a dog image and a ship images, respectively.

on labeling error and the connectivity of the augmentation graph (Equation (2) in HaoChen et al. (2021)) for the downstream linear probe error. While matching the experimental observations, they didn't offer any suggestions to reduce the dependency on labeling error. Subsequently, considering that generated data may sometimes even harm contrastive learning, Wang et al. (2024) formulated a novel analysis strategy to explore the reasons underlying this labeling error phenomenon from the perspective of data inflation [1]. Theoretical and empirical results in Wang et al. (2024) illustrated that stronger data inflation would bring larger graph connectivity, which decreases the upper bound of the downstream linear probe error. They also validated that stronger data augmentation would affect the trade-off of labeling error and graph connectivity to bring the phenomenon that model performance first rises and then falls. Consequently, they suggested obtaining large graph connectivity along with small labeling error through the combination of strong data inflation and weak augmentation. However, good downstream performance is primarily ensured by large graph connectivity. They didn't verify whether the labeling error caused by the weak augmentation is sufficiently small.

In this paper, we investigate the negative impacts of labeling error. Specifically, we first provide the theoretical upper and lower bounds of downstream classification risk and error [2] considering both intra-class and inter-class augmentation

overlap. From the data dimensionality reduction perspective, our theoretical and empirical analyses validate the effectiveness of some useful suggestions for decreasing labeling error to enhance downstream classification performance. Our main contributions are summarized as follows.

- *Theoretical guarantees of classification risk for both intra-class and inter-class augmentation overlap cases.* Beyond the intra-class augmentation overlap considered by Wang et al. (2022), we establish the first upper and lower bounds of downstream classification risk for inter-class augmentation overlap case. It is discovered from these bounds that there are several significant dependencies associated with labeling error. Following the above analysis, a new perspective of data dimensionality reduction is introduced to reduce these dependencies. As illustrated by singular value decomposition (SVD), both theoretical analysis and empirical observations demonstrate that this dimensionality reduction technique is capable of suppressing classification risk by diminishing labeling error.

- *Theoretical guarantees of classification error while using SVD.* Except for downstream classification risk, we provide an upper bound of classification error while using SVD. This bound uncovers that SVD can also reduce classification error by decreasing labeling error. Besides, SVD might lead to small graph connectivity, thereby potentially worsening this bound. We suggest that adopting a moderate [3] embedding dimension contributes to the decrease of classification error by

---

[1] Data inflation was defined by Wang et al. (2024) as the process of using generative models (e.g., denoising diffusion probabilistic model (DDPM) (Ho et al., 2020)) to generate a lot of synthetic samples. Wang et al. (2024) performed contrastive learning on the combination of the real and generated data.

[2] Classification error measures the proportion of misclassified samples. Thus, the summation of classification error and accuracy equal to 100%.

[3] The word "moderate" means that the embedding dimension $k$ is not too small or too large. In our experiments, $k = 512$ or $k = 1024$ may be suitable.

increasing graph connectivity, which is validated by our data experiments.

- *Some useful suggestions guided by our analyses.* Under the experiment setting of Wang et al. (2024), our empirical evaluations indicate that their weak data augmentation still brings an unignorable labeling error. The aforementioned analyses suggest that we should use moderate embedding dimension, data inflation, weak augmentation, and SVD to achieve two critical objectives: ensuring large graph connectivity and small labeling error, thereby enhancing the downstream classification accuracy.

## 2. Related Works

**Theoretical Understanding of Contrastive Learning.** Except for the above theoretical studies, there are also many understanding of contrastive learning from other perspectives (Tian et al., 2020; Chen et al., 2020a; Zimmermann et al., 2021; Saunshi et al., 2022; Waida et al., 2023; Ji et al., 2023; Zhang et al., 2023b; Zou & Liu, 2023; Wen et al., 2024). From the perspective of information theory, Tian et al. (2020) theoretically and empirically showed that data augmentation can decrease mutual information and improve downstream classification accuracy. Saunshi et al. (2022) empirically presented that different function classes and algorithms bring different behaviors and suggested the consideration of inductive biases in theoretical analysis. Ji et al. (2023) proved two conclusions: 1) contrastive learning outperforms some other self-supervised learning paradigms such as autoencoder and generative adversarial network; 2) label information benefits in-domain downstream task and harms out-domain downstream task. Different from previous theories on data augmentation (Dao et al., 2019; Rajput et al., 2019; Wu et al., 2020), Chen et al. (2020a) pioneered a connection between data augmentation and the performance of downstream task using group theory under approximate equality condition (similar to label consistency). These works all hinge upon the label consistency assumption as a fundamental premise. In contrast, our analysis is free from the label consistency assumption.

**Applications of Contrastive Learning.** Contrastive learning has achieve the empirical success in many fields, including time series prediction (Nonnenmacher et al., 2022; Lee et al., 2024; Xu et al., 2024; Zheng et al., 2024), graph learning (Xia et al., 2022; Ghose et al., 2023; Lin et al., 2023; Yu et al., 2023b; Liu et al., 2024), federated learning (Yu et al., 2023a; Louizos et al., 2024), multi-modal learning (Lin et al., 2022; Wang et al., 2023a; Huang et al., 2023a; Fang et al., 2023; Xia et al., 2023), adversarial learning (Xu et al., 2023a; Zhang et al., 2023a; Xu et al., 2023b; Luo et al., 2023), etc. Not only self-supervised contrastive learning, there are also some supervised works (Khosla et al., 2020; Barbano et al., 2023) and weak supervised progresses (Zheng et al., 2021; Tsai et al., 2022; Cui et al., 2023) that combine available label information to guide model training.

## 3. Preliminaries

This section initiates our analysis with a comprehensive overview of contrastive learning. We first denote $\bar{D}$ the finite but exponentially large set set of all unlabeled original sample $\bar{x} \in \mathbb{R}^d$, and denote $y : \bar{D} \to [K]$ the ground-truth (deterministic) labeling function, then $y_{\bar{x}} \in \{1, ..., K\}$. Let $\mathcal{P}$ be the population distribution of the original samples. Contrastive learning initializes and trains the model parameters in an unsupervised fashion, laying a foundational understanding of the underlying structure and patterns of original data. The subsequent supervised fine-tuning stage further adapts the pre-trained model parameters to some specific downstream task on the test data drawing from $\mathcal{P}$ to enhance model performance.

**For the first learning stage**, an encoder projector $f \in \mathcal{F}_1 : \mathbb{R}^d \to \mathbb{S}^{k-1}$ is pre-trained to map a $d$-dimensional input vector $\bar{x}$ to an embedding vector $z = f(x)$ in a $k$-dimensional unit hypersphere, where the inequality $k < d$ generally holds which is validated by our experiments. This pre-training process is composed with several steps including data augmentation, contrastive representation, and loss calculation. Due to the unavailability of true label, contrastive pre-training constructs a surrogate task via some sample augmentation strategies. Then, this task is learned by minimizing the distance between similar samples and maximizing the distance between dissimilar samples in embedding space. For example, we first select an original sample $\bar{x}$, and make data augmentations $t, t^+ \in \mathcal{T}(\mathcal{T} = \{t | t : \mathbb{R}^d \to \mathbb{R}^d\})$ to obtain two independent augmented samples $x = t(\bar{x}), x^+ = t^+(\bar{x})$, respectively. We use $p(\cdot | \bar{x})$ to denote the distribution of the augmented sample for $\bar{x}$. Then, $M$ negative samples $x_i^-, i = 1, ..., M$, are randomly augmented from other original samples. We assume $\bar{x}, x, x^+, x^-$ follow the same population distribution, which is the same as previous work like Wang et al. (2022). We use these augmented samples to calculate the most common contrastive loss, InfoNCE loss (van den Oord et al., 2018), defined as follows

$$\mathcal{L}_{InfoNCE}(f)$$
$$= \mathbb{E}_{x, x^+, \{x_i^-\}_{i=1}^M} \left[ -\log \frac{e^{f(x)^\top f(x^+)}}{e^{f(x)^\top f(x^+)} + \sum_{i=1}^M e^{f(x)^\top f(x_i^-)}} \right].$$
(1)

In InfoNCE loss, the similarity between two samples is quantified by the inner product $f(x)^\top f(x')$ of two vectors ($x'$ denotes any sample in $x^+, x_1^-, ..., x_M^-$). Considering that $f$ is a projector mapping from $\mathbb{R}^d$ to a $k$-dimensional unit hypersphere, the inner product $f(x)^\top f(x')$ represents

cosine similarity. To sum up, minimizing InfoNCE loss is equivalent to maximizing the similarity of positive pair and minimizing the similarity of negative pairs. Generally, we consider the empirical version of InfoNCE loss

$$
\hat{\mathcal{L}}_{InfoNCE}(f)
$$
$$
= -\frac{1}{n_1} \sum_{j=1}^{n_1} \log \frac{e^{f(x_j)^\top f(x_j^+)}}{e^{f(x_j)^\top f(x_j^+)} + \sum_{i=1}^{M} e^{f(x_j)^\top f(x_{ji}^-)}} \quad (2)
$$

and its minimizer

$$
f^* \in \arg\min_{f \in \mathcal{F}_1} \hat{\mathcal{L}}_{InfoNCE}(f).
$$

**For the second learning stage**, the parameters of the pre-trained encoder remain unaltered. We retrain a linear projection head $g \in \mathcal{F}_2 : \mathbb{R}^k \to \mathbb{R}^K$ with the weight $W \in \mathbb{R}^{k \times K}$ using the cross entropy (CE) loss to conduct downstream classification task. For a labeled sample $x \sim \mathcal{P}$, the CE loss and the mean CE loss [4] are calculated via

$$
\mathcal{L}_{CE}(g_{f,W}) = \mathbb{E}_x \left[ -\log \frac{\exp\left(f(x)^\top w_{y_x}\right)}{\sum_{i=1}^{K} \exp\left(f(x)^\top w_i\right)} \right] \quad (3)
$$

and

$$
\mathcal{L}_{CE}(g_{f,\mu}) = \mathbb{E}_x \left[ -\log \frac{\exp\left(f(x)^\top \mu_{y_x}\right)}{\sum_{i=1}^{K} \exp\left(f(x)^\top \mu_i\right)} \right], \quad (4)
$$

where the linear classifier predicts $g_{f,W} = \arg\max_{i \in [K]} \left(f^\top W\right)_i$, $W = [w_1, ..., w_K]$, and the parameter of mean projection head is $\mu = [\mu_1, ..., \mu_K]$ whose element $\mu_i$ denotes the mean of the representations for the inputs with the label $i \in [K]$, i.e., $\mu_i = \mathbb{E}_{\{x|y_x=i\}}[f(x)]$. Define the downstream classification error as

$$
\mathcal{E}(f, W) = \Pr_{x \sim \mathcal{P}} [g_{f,W}(x) \neq y_x]. \quad (5)
$$

This work uses linear probing in fine-tuning stage, rather than full fine-tuning which updates all model parameters.

As previously stated, contrastive learning employs data augmentation methods to make preparation for unsupervised pre-training. There are two default assumptions: 1) the positive samples pair $(x, x^+)$ have the same label, i.e., $y_x = y_{x^+}$; 2) any negative sample $x_i^-$ has a label different from $y_x$. However, the inherent randomness of both traditional data augmentations and negative sample sampling may violate these two assumptions in practice. As a consequence, the pre-training process may be led astray, potentially undermining both the overall training effectiveness

---

[4]Mean classifier was first considered by Arora et al. (2019). They stated that the mean classifier could achieve comparable performance to learned weights. Note that we don't use the mean classifier in our experiments. It is only available as an intermediate result (Theorem 4.2) in our theoretical analysis.

and the quality of the learned representations. This study primarily focuses on the potential labeling error caused by false positive augmented samples (Assumption 3.1).

**Assumption 3.1** (Labeling Error (Wang et al., 2024)). For any $\bar{x} \sim \mathcal{P}$, its latent label $y_{\bar{x}}$, and its augmented sample $x \sim p(\cdot|\bar{x})$, we assume that the true label of $x$ is not consistent with $y_{\bar{x}}$ with the probability $\alpha \in (0, 1)$. That is, $\mathbb{E}_{\bar{x} \sim \mathcal{P}, x \sim p(\cdot|\bar{x})} [\mathbb{I}[y_x \neq y_{\bar{x}}]] = \alpha$.

## 4. Main Results

### 4.1. Theoretical Impact of Labeling Error

**Definition 4.1** (Augmentation Overlap). Given a collection of augmentation strategies $\mathcal{T}$, we say that two original samples $\bar{x}, \bar{x}' \sim \mathcal{P}$ are $\mathcal{T}$-augmentation overlapped if they have overlapped views, i.e., $\exists t, t' \in \mathcal{T}$ such that $t(\bar{x}) = t'(\bar{x}')$.

In the analysis of Wang et al. (2022), they proposed the concept of augmentation overlap. Owing to the label consistency, the positive augmented sample pair $(x, x^+)$ has the same label $y_x = y_{x^+}$. Therefore, they analyzed the model performance in the case that different intra-class samples could have overlapped augmented views (Figure 1 (b)). While under Assumption 3.1, there arises the phenomenon that different inter-class samples may also exhibit overlapped augmented views (Figure 1 (c)). Our first theorem (Theorem 4.2) takes these two augmentation overlap cases into account (the proof can be found in *Appendix C*).

**Theorem 4.2** (Bounds of Mean Classification Risk). *Let Assumption 3.1 hold. For any $f \in \mathcal{F}_1, g \in \mathcal{F}_2$, the gap between the mean downstream classification risk and the contrastive risk $\mathcal{L}_{CE}(g_{f,\mu}) - \mathcal{L}_{InfoNCE}(f)$ can be upper bounded by*

$$
\sqrt{V(f(x))} + \sqrt{V^-(f(x))} + \mathcal{O}\left(M^{-\frac{1}{2}}\right) - \log\left(\frac{M}{K}\right)
$$

*and lower bounded by*

$$
-\sqrt{V(f(x))} - \sqrt{V^-(f(x))} - \frac{1}{2}V(f(x^-))
$$
$$
-\mathcal{O}\left(M^{-\frac{1}{2}}\right) - \log\left(\frac{M+1}{K}\right),
$$

*where* $V(f(x)) = \mathbb{E}_{(x,x^+) \in X^+} \left[\|f(x) - \mu_{y_x}\|^2\right]$, $V^-(f(x)) = \mathbb{E}_{(x,x^+) \in X^-} \left[\|f(x^+) - \mu_{y_x}\|^2\right]$, $V(f(x^-)) = V(z|z \in \{f(x^+), f(x^-)\}, y_{x^+} = y_x) = \mathbb{E}_{\{z|z \in \{f(x^+), f(x^-)\}, y_{x^+} = y_x\}} \left[\|z - \mu_{y_z}\|^2\right]$ [5] *are the intra-class variance of the representations for true positive augmented samples, the variance for*

---

[5]$\mu_{y_x}$ and $\mu_{y_z}$ denote the mean of these representations with the same label as sample $x$ and embedding $z$, respectively.

*false positive augmented samples, and the intra-class variance for negative and true positive augmented samples, respectively,* $X^+ = \{(x, x^+)|y_x = y_{x+}\}$ *and* $X^- = \{(x, x^+)|y_x \neq y_{x+}\}$ *denote the sets of true positive sample pair and false positive sample pair, respectively.*

From the above results, it can be observed that there are multiple terms determining the upper and lower bounds of the mean classification risk $\mathcal{L}_{CE}(g_{f,\mu})$, i.e., *1)* $\mathcal{L}_{InfoNCE}(f)$: the contrastive risk; *2)* $V(f(x))$: the intra-class variance of the representations of true positive augmented samples; *3)* $V^-(f(x))$: the variance of false positive augmented samples; *4)* $V(f(x^-)) = V(z|z \in \{f(x^+), f(x^-)\}, y_{x+} = y_x)$: the intra-class variance of negative and true positive augmented samples; *5)* $\mathcal{O}\left(M^{-\frac{1}{2}}\right)$: the order of the approximation error; *6)* $\log\left(\frac{M}{K}\right)$: a constant depending on $M$ and $K$. In terms of the form of these bounds, our result is more refined than that of Theorem 4.2 in Wang et al. (2022) as evidenced by these terms $V(f(x)), V^-(f(x))$, and $V(f(x^-))$. It should be noted that the assumption of our Theorem 4.2 is milder than the corresponding result in (Wang et al., 2022), which leads to some latent differences in these terms. We carry out the following analyses with respect to these terms.

**(1)** $V(f(x))$**:** Although this term looks like the variance term in the bound of Theorem 4.2 for Wang et al. (2022), its definition $\mathbb{E}_{(x,x^+)\in X^+}\left[\|f(x) - \mu_{y_x}\|^2\right]$ demonstrates the clustering property of true positive sample pair, not including false positive sample pair. If the value of $\alpha$ equals to zero, the variance term in Wang et al. (2022) will be equivalent to our $V(f(x))$, as shown in Figure 2 (a).

**(2)** $V^-(f(x))$**:** Wang et al. (2022) mistook false positive augmented samples for true positive augmented samples, consequently resulting in a large conditional positive intra-class variance. Upon revisiting the definition $\mu_{y_x} = \mathbb{E}_{\{x|y=y_x\}}[f(x)]$, these false positive augmented samples do not have an impact on $\mu_{y_x}$. When we discard these false positive augmented samples, the remaining positive augmented samples appear more concentrated around $\mu_{y_{\bar{x}}}$ in comparison to those of Wang et al. (2022). Therefore, as the value of $\alpha$ decreases, this term will progressively decrease until it reaches 0.

**(3)** $V(f(x^-))$**:** Considering the sampling randomness and the unavailability of the true labels for negative samples, we let the label $y^-$ denote the latent label of the negative augmented sample $x^-$, irrespective of the class to which its original sample $\bar{x}^-$ pertains. Therefore, compared with $V^-(f(x))$, this term remains unaffected by the labeling error $\alpha$. We provide Figure 2 to facilitate the understanding of the difference between the analyses of $V^-(f(x))$ and $V(f(x^-))$.

To sum up, Theorem 4.2 provides more detailed and practical bounds of the mean downstream classification risk $\mathcal{L}_{CE}(g_{f,\mu})$ under milder assumption compared to the results in Wang et al. (2022). Our analysis indicates that the labeling error $\alpha$ exerts negative impacts on the theoretical downstream classification risk. Thus, we aim to put forward some suggestions to mitigate these impacts.

### 4.2. Dimensionality Reduction as A New Perspective

This subsection investigates the impact of labeling error on contrastive learning through a new perspective of data dimensionality reduction. Specifically, we use data dimensionality reduction on individual original sample to obtain the corresponding compressed sample which is then augmented to pre-train the model. We take Singular Value Decomposition (SVD, Wall et al. (2003)) as an illustrative example. For ease of calculation, the randomized SVD (RSVD, Halko et al. (2011)) algorithm is employed in our experiments.

**Definition 4.3** (SVD)**.** For a matrix $X \in \mathbb{R}^{m \times m'}$ (without loss of generality, we let $m \leq m'$), its SVD is expressed as $X = USV^\top$, where $U = [u_1, ..., u_m] \in \mathbb{R}^{m \times m} (V = [v_1, ..., v_{m'}] \in \mathbb{R}^{m' \times m'})$ is the left (right) singular matrix, consisting of $m(m')$ orthonormal column vectors (eigenvectors of $XX^\top(X^\top X)$), $S = [diag(s_1, ..., s_m), \mathbf{0}]$ is composed of a diagonal matrix $diag(s_1, ..., s_m) \in \mathbb{R}^{m \times m}$ and a zero matrix $\mathbf{0}$ with size $m \times (m' - m)$, the values $s_i(i = 1, ..., m)$ are the singular value, arranged in descending order such that $s_1 \geq s_2 \geq ... \geq s_m \geq 0$.

In general, we use the truncated version of SVD which is expressed as $\hat{X}_q = U_q S_q V_q^\top$, where $U_q = [u_1, ..., u_q] \in \mathbb{R}^{m \times q}, S_q = diag(s_1, ..., s_q), V_q = [v_1, ..., v_q] \in \mathbb{R}^{m' \times q}, q \in [m]$. The data distribution after conducting truncated SVD is written as $\mathcal{P}_q$, then the corresponding labeling error is $\alpha_q = \mathbb{E}_{\bar{x} \sim \mathcal{P}_q, x \sim p(\cdot|\bar{x})}\left[\mathbb{I}[y_x \neq y_{\bar{x}}]\right]$. The following lemma provided by Eckart & Young (1936) proved that $\hat{X}_q$ is the best least squares lower rank approximation with rank $q$ for a given matrix $X$.

**Lemma 4.4** (Eckart-Young Theorem)**.** *Let $X$ be a $m \times m'$ matrix of rank $r \in [m]$ with complex elements. Define $P_q$ as the set of all $m \times m'$ matrices with rank $q \in [r]$. Then,*

$$\left\|X - \hat{X}_q\right\|_F \leq \|X - B\|_F, \forall B \in P_q.$$

As stated by Kilmer et al. (2021), Lemma 4.4 implies that the majority of the informational content is captured by the dominant singular subspaces, i.e., the span of the singular vectors corresponding to the largest singular values. By default, we assume a positive correlation between the amount of information and its semantic relevance. It follows that the information content corresponding to the largest singular value $s_1$ should represent the most crucial information in

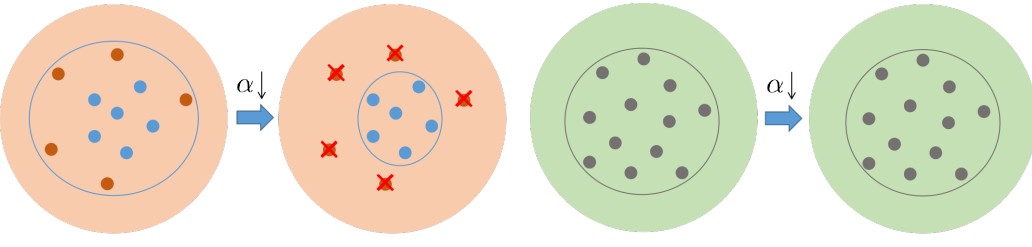

(a) Positive Augmented Samples      (b) Negative Augmented Samples

*Figure 2.* The impact of the labeling error $\alpha$ on the positive (a) and negative (b) augmented samples. Brown dots, blue dots and grey dots denote the false positive samples, true positive samples and true negative samples, respectively. Blue arrows denote the decrease of $\alpha$.

*Table 1.* Downstream classification top-1 accuracies (%) of SimCLR ($\mathcal{L}_{InfoNCE}$) on CIFAR-10 using the truncated SVD which discards two singular values ($s_{i,i+1}$ denotes we discard the $i$-th and the $i+1$-th singular values $s_i, s_{i+1}$ via SVD, $\mathcal{T}_1 = \{$Random resize crop (RRC), Color jitter, Random horizontal flip, Random grayscale, Gaussian blur$\}$, bold number indicates the optimal performance of each experimental group).

| $\mathcal{T}$ | Encoder | $s_{1,2}$ | $s_{2,3}$ | $s_{3,4}$ | $s_{4,5}$ | $s_{5,6}$ | $s_{6,7}$ | $s_{7,8}$ | $s_{8,9}$ |
|---|---|---|---|---|---|---|---|---|---|
| $\mathcal{T}_1$ | Resnet-18 | 57.31 | 64.14 | 65.63 | 67.20 | 66.93 | 67.77 | 67.93 | 68.44 |

| $\mathcal{T}$ | Encoder | $s_{9,10}$ | $s_{10,11}$ | $s_{11,12}$ | $s_{12,13}$ | $s_{15,16}$ | $s_{21,22}$ | $s_{31,32}$ | |
|---|---|---|---|---|---|---|---|---|---|
| $\mathcal{T}_1$ | Resnet-18 | 67.97 | 68.08 | 68.59 | 68.96 | 68.61 | 69.20 | **69.48** | |

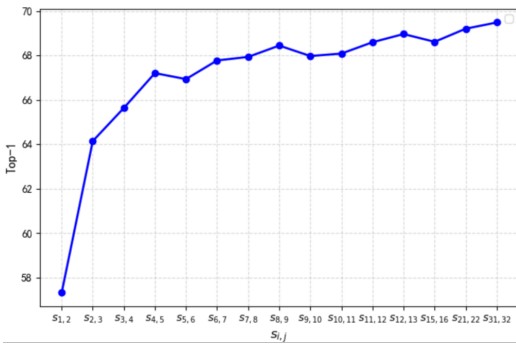

*Figure 3.* Curve of classification top-1 accuracies (%).

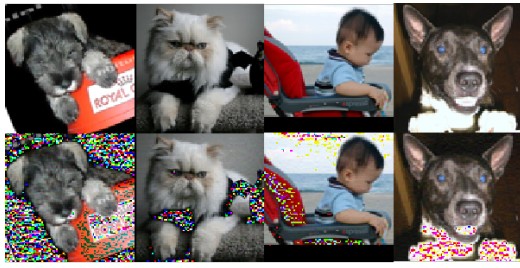

*Figure 4.* Examples from STL-10. The first row shows the original images and the second row shows the images after taking SVD.

$X$. For example, an image is intrinsically represented as a matrix $X$. The information associated with $s_1$ is most significant for distinguishing the true semantic of the image.

Table 1 and Figure 3 empirically validate that the larger the singular value is, the more semantic-related information it encompasses. Figure 4 visually presents the differences between original samples and the compressed samples.

**Assumption 4.5.** Let a sample and its corresponding sample after applying SVD be represented as the matrices $X$ and $\hat{X}_q \in \mathbb{R}^{m \times m'}$, respectively. Assume that there are $q^*$ singular values associated with semantic-related information. When $q \geq q^*$, under Assumption 3.1 and the augmentation collection $\mathcal{T}$, the true label of the augmented sample of $\hat{X}_q$ is inconsistent with the latent label of $X$ with the probability $\alpha_q \leq \alpha$. Conversely, when $q < q^*$, the corresponding probability satisfies $\alpha_q > \alpha_{q^*}$.

The truncated SVD with $q = q^*$ is defined as the optimal truncated SVD. As indicated by Assumption 4.5, we assume the trend where $\alpha$ initially decreases and then increases as the decrease of $q$. In the first stage ($q \geq q^*$), the decrease of $\alpha$ derives from the removal of semantically irrelevant information. Once $q < q^*$, semantic-related information begins to be discarded, giving rise to the increase of $\alpha$. Table 2 and Figure 5 (*Appendix G*) present empirical performances applying the truncated SVD with different values of $q$, aligning with our analysis. Notably, the values of $q^*$ vary across different settings. Table 3 further validates the effectiveness of SVD across various augmentation strategies [6].

---

[6]Explanations of these augmentation strategies in Table 3 are provided in *Appendix D*

*Table 2.* Downstream classification top-1 accuracies (%) of SimCLR ($\mathcal{L}_{InfoNCE}$) using the truncated SVD with different truncated parameter $q$.

| $\mathcal{T}$ | Encoder | Dataset | w/o SVD | $q = 30$ | $q = 25$ | $q = 20$ | $q = 15$ | $q = 10$ |
|---|---|---|---|---|---|---|---|---|
| $\mathcal{T}_1$ | Resnet-18 | CIFAR-10 | 68.82 | 69.48 | 69.75 | **69.87** | 69.01 | 68.26 |
| $\mathcal{T}_1$ | Resnet-50 | CIFAR-10 | 63.20 | 63.36 | **63.96** | 62.23 | 60.97 | 60.06 |
| RRC | Resnet-18 | CIFAR-10 | 58.56 | **58.83** | 58.67 | 58.61 | 58.54 | 58.32 |
| $\mathcal{T}_1$ | Resnet-18 | CIFAR-100 | 38.48 | 38.81 | **40.10** | 39.05 | 38.98 | 38.10 |
| $\mathcal{T}$ | Encoder | Dataset | w/o SVD | $q = 90$ | $q = 70$ | $q = 50$ | $q = 30$ | $q = 10$ |
| $\mathcal{T}_1$ | Resnet-18 | STL-10 | 71.54 | **73.12** | 72.29 | 71.10 | 70.04 | 67.52 |

*Table 3.* Downstream classification top-1 accuracies (%) of SimCLR ($\mathcal{L}_{InfoNCE}$) on CIFAR-10 using the truncated SVD with different augmentations ($\mathcal{T}_2 = \{\mathcal{T}_1 +$ Cutout$\}$; $\mathcal{T}_3 = \{$RRC, Cutout, Hide patch$\}$; $\mathcal{T}_4 = \{$RRC, Cutout, Color jitter$\}$; $\mathcal{T}_5 = \{$RRC, Cutout$\}$; $\mathcal{T}_6 = \{$RRC(0.08, 0.5), Cutout$\}$; $\mathcal{T}_7 = \{$RRC(0.08, 0.5), Cutout(0.5, 1.0)$\}$).

| SVD | Encoder | $\mathcal{T}_2$ | $\mathcal{T}_3$ | $\mathcal{T}_4$ | $\mathcal{T}_5$ | $\mathcal{T}_6$ | $\mathcal{T}_7$ | RRC(0.08,0.5) |
|---|---|---|---|---|---|---|---|---|
| w.o. SVD | Resnet-18 | 62.90 | 50.53 | 60.00 | 56.67 | 54.97 | 54.09 | 57.11 |
| $q = 30$ | Resnet-18 | **64.86** | **51.00** | **61.57** | **57.85** | **55.69** | **54.75** | **58.10** |

Due to the unavailability of $q^*$, we select a relatively large value of $q$, specifically $q = 30$ for CIFAR-10 (image size is $32 \times 32$) in Table 3. Even though we merely discard the information corresponding to the two smallest singular values $s_{31}, s_{32}$, the results of multiple augmentation strategies in Table 3 all exhibit some non-negligible improvements, with an average increase of 0.97%.

Further, we introduce an assumption related to SVD, i.e., $(\epsilon_{q^*}, \epsilon_q)$-alignment for any false positive sample pair.

**Assumption 4.6** (($\epsilon_{q^*}, \epsilon_q$)-Alignment for Any False Positive Sample Pair). *Let Assumptions 3.1, 4.5 hold and $X^- = \{(x, x^+)|y_x \neq y_{x^+}\}$. When performing SVD with the truncated value $q$, the encoder $f$ with the empirical InfoNCE loss $\hat{\mathcal{L}}_{InfoNCE}(f)$ (2) can align any false positive sample pair $(x, x^+) \in X^-$, such that their distance in the embedding space lies within $[\epsilon(\alpha_{q^*}), \epsilon(\alpha_q)]$, i.e., $\epsilon(\alpha_{q^*}) = \min_{(x,x^+)\in X^-} \|f(x) - f(x^+)\|$ and $\epsilon(\alpha_q) = \max_{(x,x^+)\in X^-} \|f(x) - f(x^+)\|$. For simplicity, let $\epsilon_{q^*} = \epsilon(\alpha_{q^*}), \epsilon_q = \epsilon(\alpha_q)$.*

Intuitively, labeling error can affect the relationship between the embedding vectors of a positive augmented sample pair. Thus, we introduce the $(\epsilon_{q^*}, \epsilon_q)$-alignment assumption, which is inspired by the weak alignment definition proposed by Wang et al. (2022) (Definition B.1 in *Appendix B.1*). Furthermore, Assumption 4.6 means that the encoder $f$, optimized under the loss $\hat{\mathcal{L}}_{InfoNCE}(f)$, can align any $(x, x^+) \in \{(x, x^+)|y_x = y_{x^+}\}$ with the distance at most $\epsilon_{q^*}$ in embedding space, that is, $\max_{(x,x^+)\in X^+} \|f(x) -$

$f(x^+)\| \leq \epsilon_{q^*}$.

Theorem 4.7, stated as below, is derived based on Assumption 4.6, which theoretically clarifies the effectiveness of data dimensionality reduction.

**Theorem 4.7.** *Given the conditions of Theorem 4.2 and Assumption 4.6, after taking the truncated SVD on $\bar{\mathcal{D}}$, the mean downstream classification risk $\mathcal{L}_{CE}(g_{f,\mu}) - \mathcal{L}_{InfoNCE}(f)$ with the encoder $f$ can be upper bounded by*

$$\epsilon_{q^*} + \epsilon_q + \mathcal{O}\left(M^{-\frac{1}{2}}\right) - \log\left(\frac{M}{K}\right)$$

*and lower bounded by*

$$-\epsilon_{q^*} - \epsilon_q - \frac{1}{2}V(f(x^-)) - \mathcal{O}\left(M^{-\frac{1}{2}}\right) - \log\left(\frac{M+1}{K}\right).$$

*When $q = q^*$, the two bounds are $\epsilon_{q^*} + \mathcal{O}\left(M^{-\frac{1}{2}}\right) - \log\left(\frac{M}{K}\right)$ and $-\epsilon_{q^*} - V(f(x^-)) - \mathcal{O}\left(M^{-\frac{1}{2}}\right) - \log\left(\frac{M+1}{K}\right)$.*

### 4.3. Further Understanding of Labeling Error

Next, we will delve deeper into the theoretical impact of SVD on the downstream classification error $\mathcal{E}(f, W)$ in (5) through the lens of spectral graph theory. In the subsequent analysis, the spectral contrastive loss $\mathcal{L}_{spe}(f)$ is adopted,

$$\mathcal{L}_{spe}(f)$$
$$= -2\mathbb{E}_{x,x^+}\left[f(x)^\top f(x^+)\right] + \mathbb{E}_{x,x^-}\left[\left(f(x)^\top f(x^-)\right)^2\right],$$

*Table 4.* Downstream classification top-1 accuracies (%) of SimCLR ($\mathcal{L}_{spe}$) on CIFAR-10 using the truncated SVD with different $q$ or the data inflation strategy under the weak data augmentation adopted by Wang et al. (2024) ($\mathcal{T}_8 = \{$RRC(0.2, 1.0), Color jitter(0.5, 0.4), Random horizontal flip, Random grayscale, Gaussian blur$\}$).

| $\mathcal{T}$ | Encoder | Inflation | w/o SVD | $q = 30$ | $q = 25$ | $q = 20$ | $q = 15$ | $q = 10$ |
|---|---|---|---|---|---|---|---|---|
| $\mathcal{T}_8$ | Resnet-18 | 71.54 | 71.21 | 71.64 | **71.65** | 71.11 | 70.41 | 67.83 |

| $\mathcal{T}$ | Encoder | Inflation | Inflation + ($q = 30$) | | Inflation + ($q = 25$) | | Inflation + ($q = 20$) | |
|---|---|---|---|---|---|---|---|---|
| $\mathcal{T}_8$ | Resnet-18 | 71.54 | 71.64 | | **72.55** | | 71.19 | |

*Table 5.* Downstream classification top-1 accuracies (%) of SimCLR ($\mathcal{L}_{InfoNCE}$) on CIFAR-10 using the truncated SVD with different $q$ or the data inflation strategy under the weak data augmentation adopted by Wang et al. (2024) ($\mathcal{T}_8 = \{$RRC(0.2, 1.0), Color jitter(0.5, 0.4), Random horizontal flip, Random grayscale, Gaussian blur$\}$).

| $\mathcal{T}$ | Encoder | Inflation | w/o SVD | $q = 30$ | $q = 25$ | $q = 20$ | $q = 15$ | $q = 10$ |
|---|---|---|---|---|---|---|---|---|
| $\mathcal{T}_8$ | Resnet-18 | 70.87 | 70.11 | 70.92 | 70.67 | **70.97** | 70.41 | 69.03 |

| $\mathcal{T}$ | Encoder | Inflation | Inflation + ($q = 30$) | | Inflation + ($q = 25$) | | Inflation + ($q = 20$) | |
|---|---|---|---|---|---|---|---|---|
| $\mathcal{T}_8$ | Resnet-18 | 70.87 | 70.98 | | **71.21** | | 70.48 | |

which is proposed in HaoChen et al. (2021) and similar to the InfoNCE loss. The augmentation graph, defined as below, is involved in our analysis.

**Definition 4.8** (Augmentation Graph, HaoChen et al. (2021)). Given an augmentation collection $\mathcal{T}$, there exist $n$ augmented samples that form the augmentation dataset

$$\mathcal{D}_{aug} = \{x | x = t(\bar{x}), \bar{x} \sim \mathcal{P}, t \in \mathcal{T}\}.$$

An augmentation graph $\mathcal{G}$ is obtained by taking the $n$ augmented samples as the graph vertices and assuming there exists an edge between two vertices $x, x' \in \mathcal{D}_{aug}$ (if they can be generated from a random original sample $\bar{x} \sim \mathcal{P}$).

According to spectral graph theory, we define $A \in \mathbb{R}^{n \times n}$ as the adjacency matrix of the augmentation graph $\mathcal{G}$. For two augmented samples $x, x' \in \mathcal{D}_{aug}$, the element $A(x, x')$ denotes the marginal probability of generating $x, x'$ from a random original sample $\bar{x} \sim \mathcal{P}$. Formally, $A(x, x') = \mathbb{E}_{\bar{x} \sim \mathcal{P}} [p(x|\bar{x})p(x'|\bar{x})]$. The corresponding normalized graph Laplacian matrix is $L = I - D^{-\frac{1}{2}} A D^{-\frac{1}{2}}$, where $D$ denotes a diagonal degree matrix with the diagonal element $D_{x,x} = \sum_{x' \in \mathcal{D}_{aug}} A(x, x')$. The eigenvalues of $L$ are denoted as $\{\lambda_i\}_{i=1}^n$, where $0 = \lambda_1 \leq ... \leq \lambda_n \leq 2$. [7]

HaoChen et al. (2021) first established an upper bound related to $\alpha$ and $\lambda_{k+1}$ for the downstream classification error of the majority voting classifier $\bar{g}_{f,W}(\bar{x}) = \arg\max_{i \in [K]} \Pr_{x \sim p(\cdot|\bar{x})} [g_{f,W}(x) = i]$. Building upon their

analysis framework, Wang et al. (2024) suggested using strong data inflation and weak data augmentation to guarantee a small value of $\alpha$ and a large value of $\lambda_{k+1}$. We aim to replace the complex data inflation with SVD.

Our experimental results in Table 4 and Table 5 demonstrate that the benefit of employing SVD can catch up with that of data inflation. Moreover, we theoretically provides the upper bound of the downstream classification error under the application of the truncated SVD with the hyper-parameter $q$.

**Theorem 4.9** (Bounds of Classification Error). *Let Assumption 3.1 hold. For the empirical optimal encoder $f^*$ taking the truncated SVD with hyper-parameter $q$ on unlabeled original dataset, there exists a linear head $W^* \in \mathbb{R}^{k \times K}$ with norm $\|W^*\|_F \leq 1/(1 - \lambda_{k,q})$ such that*

$$\mathcal{E}(f^*, W^*) \leq \frac{4\alpha_q}{\lambda_{k+1,q}} + 8\alpha_q,$$

*where $k$ denotes the dimension of embedding space and $\lambda_{k+1,q}$ denotes the $k + 1$-th eigenvalues of $L_q$.*

Note that there are two differences between our Theorem 4.9 and Theorem C.3 of HaoChen et al. (2021): 1) we take the truncated SVD on original samples; 2) we evaluate the classification error for the linear classifier $g_{f,W}$ rather than the majority voting classifier $\bar{g}$ of HaoChen et al. (2021).

Wang et al. (2024) concluded that stronger data inflation solely enhances graph connectivity (larger $\lambda_{k+1}$) without affecting labeling error. When the graph connectivity is sufficient, only weak data augmentation is necessary to

---

[7]When taking the truncated SVD with hyperparameter $q$ on the unlabeled original sample $\bar{x}$, the corresponding symbols are $\mathcal{G}_q, A_q, L_q$, and $\lambda_{i,q}, i \in [n]$.

*Table 6.* Downstream classification top-1 accuracies (%) of SimCLR ($\mathcal{L}_{spe}$) using the truncated SVD ($q = 30$ for CIFAR-10 and CIFAR-100, $q = 90$ for STL-10) with different embedding dimension $k$.

| $\mathcal{T}$ | Encoder | Dataset | Embedding Dimension | | | | |
|---|---|---|---|---|---|---|---|
| | | | $k = 128$ | $k = 256$ | $k = 512$ | $k = 1024$ | $k = 2048$ |
| $\mathcal{T}_1$ | Resnet-18 | CIFAR-10 | 67.71 | 68.51 | 68.54 | **69.09** | 68.65 |
| $\mathcal{T}_1$ | Resnet-50 | CIFAR-10 | **67.43** | 65.99 | 66.50 | 66.83 | 66.22 |
| $\mathcal{T}_1$ | Resnet-18 | CIFAR-100 | 35.00 | 36.68 | 36.78 | **37.78** | 37.18 |
| $\mathcal{T}_1$ | Resnet-50 | CIFAR-100 | 35.46 | 35.42 | 35.39 | **35.59** | 35.53 |
| $\mathcal{T}_1$ | Resnet-18 | STL-10 | 72.35 | 72.42 | 73.12 | **73.88** | 73.47 |
| $\mathcal{T}_1$ | Resnet-50 | STL-10 | 74.68 | 74.94 | 75.01 | **76.26** | 75.57 |

achieve a small labeling error. However, Table 4 and Table 5 reveal that the weak augmentation employed by Wang et al. (2024) still induces an non-negligible labeling error. Compared with Theorem 4.1 in Wang et al. (2024), Theorem 4.9 further introduces SVD to offer an improvement by reducing $\alpha$ to $\alpha_q$. In the following, we will analyze the impact of SVD on $\alpha_q$ and $\lambda_{k+1,q}$ involved in Theorem 4.9.

**(1) $\alpha_q$:** As discussed behind Assumption 4.5, $\alpha_q$ exhibits a trend of first decreasing and then increasing as the decrease of $q \in [m]$. The value of $\alpha_q$ reaches its optimal value $\alpha_{q^*}$ when $q = q^*$.

**(2) $\lambda_{k+1,q}$:** As mentioned earlier, $\{\lambda_i\}_{i=1}^n$ are the eigenvalues of the normalized graph Laplacian matrix $L$. Based on the definition of $L$, it can be known that $\{1 - \lambda_i\}_{i=1}^n$ are the eigenvalues of the adjacency matrix $A$, where $1 - \lambda_{k+1}$ is the $k + 1$-th largest eigenvalue. The definition of $A(x, x')$ demonstrates that the implementation of SVD on original samples will not lead to a decrease in the elements on the main diagonal of $A$. In other words, the trace $\text{tr}(A)$ is non-decreasing and remains less than 1, formally, $\text{tr}(A) = \sum_{i=1}^n (1 - \lambda_{i,q}) \leq 1$. Therefore, the application of SVD may lead to $\lambda_{k+1,q} \leq \lambda_{k+1}$, deteriorating the bound of Theorem 4.9. It appears that choosing a large dimension $k$ of embedding space may alleviate this deterioration as much as possible by increasing graph connectivity. When the value of $k$ is not sufficiently large, such as $k \leq 512$, the corresponding results in Table 6 may reveal this trend. However, Table 6 also indicates that an overly large value of $k$ might cause the degeneration of the downstream classification performance. Interestingly, similar optimal values of $k$ have been observed across experiments on different datasets (CIFAR-10, CIFAR-100 and STL-10) and different encoders (Resnet-18 and Resnet-50). Experimental results with InfoNCE loss (Table 8 in *Appendix G*) also reveal this phenomenon. Therefore, there are still some results that do not align with our analysis. As stated in Limitation (*Appendix F*), we leave the exploration of the underlying

reasons for future work. In this paper, we propose selecting a moderate value of $k$, such as 512 or 1024. Additionally, we recommend adopting data inflation similar to that in Wang et al. (2024) to mitigate the negative impact of SVD on $\lambda$, which is validated by the experiments in the second rows of Table 4 and Table 5.

In summary, an augmentation strategy is proposed and validated to ensure large graph connectivity and small labeling error, where the key building blocks include using moderate embedding dimension, data inflation, weak augmentation and SVD.

## 5. Conclusions

This paper investigated theoretically the impact of labeling error on the downstream classification performance of contrastive learning. The derived upper and lower bounds of the downstream classification risk are both affected by the labeling error. To mitigate these negative impacts, we first propose removing the semantically irrelevant information of the original data from the perspective of data dimensionality reduction. Specifically, the classical SVD method is employed to offer both theoretical and empirical evidence to support the effectiveness of dimensionality reduction. Except for the advantages of conducting SVD on the original data, we also theoretically find that SVD may cause the deterioration of downstream classification accuracy by decreasing the graph connectivity of augmentation graph. Based on the aforementioned analysis, we provide an augmentation strategy that we should use moderate embedding dimension (such as $k = 512, 1024$), data inflation, weak augmentation and SVD to ensure large graph connectivity and small labeling error, ultimately improving model performance.

## Impact Statement

This paper presents work whose goal is to advance the field of Machine Learning. There are many potential societal consequences of our work, none which we feel must be specifically highlighted here.

## Acknowledgments

This work is supported by the National Natural Science Foundation of China (NSFC) (Nos. 62376104 and 12426512) and the Open Research Fund of Engineering Research Center of Intelligent Technology for Agriculture, Ministry of Education (No. ERCITA-KF002).

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

## A. Notations

The main notations of this paper are summarized in Table 7.

*Table 7.* Summary of main notations involved in this paper.

| Notations | Descriptions |
|---|---|
| $\bar{\mathcal{D}}$ | the original unlabeled dataset drawing from the distribution $\mathcal{P}$ |
| $p(x|\bar{x})$ | the distribution of the augmented sample $x$ conditioned on the unlabeled original sample $\bar{x}$ |
| $p(x, x^+)$ | the joint distribution of the positive augmented sample pair $(x, x^+)$ |
| $\{x_i^-\}_{i=1}^M$ | $M$ negative augmented samples |
| $\mathcal{F}_1, \mathcal{F}_2$ | the functional spaces of the encoder $f$ and the linear projection head $g$ |
| $k$ | the dimension of embedding vector |
| $K$ | the number of labels for downstream classification task |
| $\mathcal{T}$ | the data augmentation set defined as $\{t|t : \mathbb{R}^d \to \mathbb{R}^d\}$ |
| $\mathcal{L}_{InfoNCE}(f)$ | population InfoNCE loss |
| $\hat{\mathcal{L}}_{InfoNCE}(f)$ | empirical InfoNCE loss |
| $f^*$ | the empirical optimal encoder for $\min_{f \in \mathcal{F}_1} \hat{\mathcal{L}}_{InfoNCE}(f)$ |
| $\mathcal{L}_{CE}(g)$ | cross entropy (CE) loss |
| $g_{f,W}, g_{f,\mu}$ | linear classifier and mean classifier |
| $W$ | the weight of the linear projection head $g$ defined as $[w_1, ..., w_K]$ |
| $\mu$ | the parameter of mean projection head defined as $[\mu_1, ..., \mu_K]$ |
| $\mathcal{E}(f, W)$ | the downstream classification error defined as $\Pr_{x \sim \mathcal{P}}[g_{f,W}(x) \neq y_x]$ |
| $\alpha$ | the labeling error defined as $\mathbb{E}_{\bar{x} \sim \mathcal{P}, x \sim p(\cdot|\bar{x})}[\mathbb{I}[y_x \neq y_{\bar{x}}]]$ |
| $m, m'$ | the size of the matrix $X \in \mathbb{R}^{m \times m'}$ (for example an image) |
| $U(V)$ | the left (right) singular matrix with $m(m')$ orthonormal column vectors |
| $S$ | the diagonal matrix whose diagonal elements are singular values |
| $q, q^*$ | the truncated parameter of truncated SVD and its optimal value |
| $\mathcal{L}_{spe}$ | the spectral contrastive loss |
| $\mathcal{D}_{aug}, n$ | the augmentation dataset defined as $\{x|x = t(\bar{x}), \bar{x} \in \bar{\mathcal{D}}, t \in \mathcal{T}\}$ and its number |
| $\mathcal{G}, A, L$ | augmentation graph, its adjacency matrix and normalized graph Laplacian matrix |
| $\lambda$ | the eigenvalues of $L$ |

## B. Lemmas

**Lemma B.1** (Lemma A.2 in Wang et al. (2022))**.** *For* $\text{LSE} := \log \mathbb{E}_{p(z)}\left[\exp\left(f(x)^\top g(z)\right)\right]$, *we denote its (biased) Monte Carlo estimate with $M$ random samples $z_i \sim p(z), i = 1, ..., M$ as* $\hat{\text{LSE}}_M = \log \frac{1}{M} \sum_{i=1}^M \exp\left(f(x)^\top g(z_i)\right)$. *Then, approximation error can be upper bounded in expectation as*

$$\mathbb{E}_{x,z_i}\left[\left|\hat{\text{LSE}}_M - \text{LSE}\right|\right] \leq \mathcal{O}\left(M^{-\frac{1}{2}}\right).$$

**Lemma B.2** (Equation (11) in Wang et al. (2022))**.** *Let a projector $f \in \mathcal{F} : \mathbb{R}^d \to \mathbb{S}^{k-1}$ map from the $d$-dimensional input space to a unit hypersphere in the $k$-dimensional space. For $x, x^+ \in \mathbb{R}^d$, we have*

$$f(x)\left(f(x^+) - \mu_y\right) \leq \left(\frac{f(x^+) - \mu_y}{\|f(x^+) - \mu_y\|}\right)^\top \left(f(x^+) - \mu_y\right) = \left\|f(x^+) - \mu_y\right\|,$$

*where $\mu_y = \mathbb{E}_{p(x|y)}[f(x)]$, $y$ denotes label.*

**Lemma B.3** (Corollary 3.5 in Budimir et al. (2000))**.** *Let the function $g : \mathbb{R}^d \to \mathbb{R}$ be a differentiable convex and $L$-smooth mapping. Then, for any $z \in \mathbb{R}^d$, we have*

$$0 \leq \mathbb{E}_{p(z)}\left[g(z)\right] - g(\mathbb{E}_{p(z)}\left[z\right]) \leq L \sum_{j=1}^{d} V(z^{(j)}) = LV(z),$$

*where $z^{(j)}$ denotes the $j$-th dimension of $z$, $V(z^{(j)})$ denotes the variance of $z^{(j)}$.*

**Lemma B.4** (Theorem 4.5 in Arora et al. (2019))**.** *With probability at least $1 - \delta$, for any $f^* \in \arg\min_{f \in \mathcal{F}_1} \hat{\mathcal{L}}_{InfoNCE}(f)$ and $g \in \mathcal{F}_2$, there holds that $\mathcal{L}_{CE}(g_{f^*, W}) \leq \mathcal{L}_{CE}(g_{f^*, \mu})$.*

**Lemma B.5** (Theorem C.3 in HaoChen et al. (2021))**.** *Assume the set of augmented data $\mathcal{D}_{aug}$ is finite. Let $f^* \in \arg\min_{f:\mathcal{D}_{aug} \to \mathbb{R}^k}$ be a minimizer of the population spectral contrastive loss $\mathcal{L}_{spe}(f)$ with $k \in \mathcal{Z}^+$. Then, there exists a linear head $W^* \in \mathbb{R}^{k \times K}$ with norm $\|W^*\|_F \leq 1/(1 - \lambda_k)$ such that*

$$\Pr_{\bar{x} \sim \mathcal{P}, x \sim p(\cdot|\bar{x})}\left[g_{f^*, W^*}(x) \neq y_{\bar{x}}\right] \leq \frac{4\alpha}{\lambda_{k+1}} + 8\alpha.$$

## C. Proofs of main results

**Theorem C.1** (Theorem 4.2 (restated))**.** *Let Assumption 3.1 hold. For any $f \in \mathcal{F}_1, g \in \mathcal{F}_2$, the gap between the mean downstream classification risk and the contrastive risk $\mathcal{L}_{CE}(g_{f,\mu}) - \mathcal{L}_{InfoNCE}(f)$ can be upper bounded by*

$$\sqrt{V\left(f(x)\right)} + \sqrt{V^-\left(f(x)\right)} + \mathcal{O}\left(M^{-\frac{1}{2}}\right) - \log\left(\frac{M}{K}\right)$$

*and lower bounded by*

$$-\sqrt{V\left(f(x)\right)} - \sqrt{V^-\left(f(x)\right)} - \frac{1}{2}V(f(x^-)) - \mathcal{O}\left(M^{-\frac{1}{2}}\right) - \log\left(\frac{M+1}{K}\right),$$

*where $V\left(f(x)\right) = \mathbb{E}_{(x,x^+)\in X^+}\left[\|f(x) - \mu_{y_x}\|^2\right]$, $V^-\left(f(x)\right) = \mathbb{E}_{(x,x^+)\in X^-}\left[\|f(x^+) - \mu_{y_x}\|^2\right]$, $V(f(x^-)) = V(z|z \in \{f(x^+), f(x^-)\}, y_{x^+} = y_x) = \mathbb{E}_{\{z|z\in\{f(x^+),f(x^-)\}, y_{x^+}=y_x\}}\left[\|z - \mu_{y_z}\|^2\right]$ are the intra-class variance of the representations for true positive augmented samples, the variance for false positive augmented samples, and the intra-class variance for negative and true positive augmented samples, respectively.*

**Proof of Theorem 4.2**: Assume positive augmented samples are $x, x^+$ and $M$ negative samples are $x_i^-, i = 1, ..., M$ ($x_i^-$ belongs to any class in the $K$ classes). We let $\mu_{y_x} = \mathbb{E}_{\{x|y=y_x\}}[f(x)]$. Under our assumptions, the labels of $x$ and $x^+$ may be different from each other. In other words, data augmentation may change the semantics of the original samples due to the intrinsic randomness. Thus, we denote the sets of true positive sample pair and false positive sample pair as $X^+ = \{(x, x^+)|y_x = y_{x^+}\}$ and $X^- = \{(x, x^+)|y_x \neq y_{x^+}\}$, respectively. Then, we have the following lower bounds of

the InfoNCE loss

$$\mathcal{L}_{InfoNCE}(f)$$

$$=\mathbb{E}_{x,x^+,\{x_i^-\}_{i=1}^M}\left[-\log\frac{e^{f(x)^\top f(x^+)}}{e^{f(x)^\top f(x^+)}+\sum_{i=1}^M e^{f(x)^\top f(x_i^-)}}\right]$$

$$=\mathbb{E}_{x,x^+,\{x_i^-\}_{i=1}^M}\left[\log\left(1+\frac{\sum_{i=1}^M\exp\left(f(x)^\top f(x_i^-)\right)}{\exp\left(f(x)^\top f(x^+)\right)}\right)\right]$$

$$\geq-\mathbb{E}_{x,x^+}\left[f(x)^\top f(x^+)\right]+\mathbb{E}_{x,\{x_i^-\}_{i=1}^M}\left[\log\sum_{i=1}^M\exp\left(f(x)^\top f(x_i^-)\right)\right]$$

$$=-\mathbb{E}_{x,x^+}\left[f(x)^\top f(x^+)\right]+\mathbb{E}_{x,\{x_i^-\}_{i=1}^M}\left[\log\frac{1}{M}\sum_{i=1}^M\exp\left(f(x)^\top f(x_i^-)\right)\right]+\log M$$

$$\overset{(1)}{\geq}-\mathbb{E}_{x,x^+}\left[f(x)^\top f(x^+)\right]+\mathbb{E}_x\left[\log\frac{1}{M}\mathbb{E}_{\{x_i^-\}_{i=1}^M}\left[\sum_{i=1}^M\exp\left(f(x)^\top f(x_i^-)\right)\right]\right]-\mathcal{O}\left(M^{-\frac{1}{2}}\right)+\log M$$

$$=-\mathbb{E}_{x,x^+}\left[f(x)^\top f(x^+)\right]+\mathbb{E}_x\left[\log\mathbb{E}_{x^-}\left[\exp\left(f(x)^\top f(x^-)\right)\right]\right]-\mathcal{O}\left(M^{-\frac{1}{2}}\right)+\log M$$

$$=-\mathbb{E}_{(x,x^+)\in X^+}\left[f(x)^\top f(x^+)\right]-\mathbb{E}_{(x,x^+)\in X^-}\left[f(x)^\top f(x^+)\right]$$
$$+\mathbb{E}_x\left[\log\mathbb{E}_{y^-}\left[\mathbb{E}_{\{x^-|y_{x^-}=y^-\}}\left[\exp\left(f(x)^\top f(x^-)\right)\right]\right]\right]-\mathcal{O}\left(M^{-\frac{1}{2}}\right)+\log M$$

$$\overset{(2)}{\geq}-\mathbb{E}_{(x,x^+)\in X^+}\left[f(x)^\top\left(\mu_{y_x}+f(x^+)-\mu_{y_x}\right)\right]-\mathbb{E}_{(x,x^+)\in X^-}\left[f(x)^\top\left(\mu_{y_x}+f(x^+)-\mu_{y_x}\right)\right]$$
$$+\mathbb{E}_x\left[\log\mathbb{E}_{y^-}\left[\exp\left(f(x)^\top\mu_{y^-}\right)\right]\right]-\mathcal{O}\left(M^{-\frac{1}{2}}\right)+\log M$$

$$=-\mathbb{E}_{(x,x^+)\in X^+}\left[f(x)^\top\mu_{y_x}+f(x)^\top\left(f(x^+)-\mu_{y_x}\right)\right]-\mathbb{E}_{(x,x^+)\in X^-}\left[f(x)^\top\mu_{y_x}+f(x)^\top\left(f(x^+)-\mu_{y_x}\right)\right]$$
$$+\mathbb{E}_x\left[\log\mathbb{E}_{y^-}\left[\exp\left(f(x)^\top\mu_{y^-}\right)\right]\right]-\mathcal{O}\left(M^{-\frac{1}{2}}\right)+\log M$$

$$\overset{(3)}{\geq}-\mathbb{E}_{(x,x^+)\in X^+}\left[f(x)^\top\mu_{y_x}+\|f(x^+)-\mu_{y_x}\|\right]+\mathbb{E}_x\left[\log\mathbb{E}_{y^-}\left[\exp\left(f(x)^\top\mu_{y^-}\right)\right]\right]$$
$$-\mathcal{O}\left(M^{-\frac{1}{2}}\right)+\log M-\mathbb{E}_{(x,x^+)\in X^-}\left[f(x)^\top\mu_{y_x}+\|f(x^+)-\mu_{y_x}\|\right]$$

$$\overset{(4)}{\geq}-\mathbb{E}_{(x,x^+)\in X^+}\left[f(x)^\top\mu_{y_x}\right]-\sqrt{\mathbb{E}_{(x,x^+)\in X^+}\left[\|f(x)-\mu_{y_x}\|^2\right]}+\mathbb{E}_x\left[\log\frac{1}{K}\sum_{k=1}^K\exp\left(f(x)^\top\mu_k\right)\right]$$
$$-\mathcal{O}\left(M^{-\frac{1}{2}}\right)+\log M-\mathbb{E}_{(x,x^+)\in X^-}\left[f(x)^\top\mu_{y_x}\right]-\sqrt{\mathbb{E}_{(x,x^+)\in X^-}\left[\|f(x^+)-\mu_{y_x}\|^2\right]}$$

$$=-\mathbb{E}_x\left[f(x)^\top\mu_{y_x}-\log\sum_{k=1}^K\exp\left(f(x)^\top\mu_k\right)\right]-\sqrt{\mathbb{E}_{(x,x^+)\in X^+}\left[\|f(x)-\mu_{y_x}\|^2\right]}$$
$$-\mathcal{O}\left(M^{-\frac{1}{2}}\right)+\log\left(\frac{M}{K}\right)-\sqrt{\mathbb{E}_{(x,x^+)\in X^-}\left[\|f(x^+)-\mu_{y_x}\|^2\right]}$$

$$=\mathcal{L}_{CE}(g_{f,\mu})-\sqrt{\mathbb{E}_{(x,x^+)\in X^+}\left[\|f(x)-\mu_{y_x}\|^2\right]}-\mathcal{O}\left(M^{-\frac{1}{2}}\right)+\log\left(\frac{M}{K}\right)-\sqrt{\mathbb{E}_{(x,x^+)\in X^-}\left[\|f(x^+)-\mu_{y_x}\|^2\right]},$$

where inequality (1) derives from Lemma B.1, inequality (2) is due to the Jensen's inequality of convex function (exponential function $\exp$), inequality (3) follows Lemma B.2, and inequality (4) is from the Cauchy–Schwarz inequality. Let $V(f(x))=\mathbb{E}_{(x,x^+)\in X^+}\left[\|f(x)-\mu_{y_x}\|^2\right]$ and $V^-(f(x))=\mathbb{E}_{(x,x^+)\in X^-}\left[\|f(x^+)-\mu_{y_x}\|^2\right]$. We can get the upper bound of $\mathcal{L}_{CE}(g_{f,\mu})+\log\left(\frac{M}{K}\right)-\mathcal{L}_{InfoNCE}(f)$ as

$$\mathcal{L}_{CE}(g_{f,\mu})+\log\left(\frac{M}{K}\right)-\mathcal{L}_{InfoNCE}(f)\leq\sqrt{V^-(f(x))}+\sqrt{V(f(x))}+\mathcal{O}\left(M^{-\frac{1}{2}}\right).$$

Next, we will prove the corresponding lower bound.

$$\mathcal{L}_{CE}(g_{f,\mu})$$

$$= - \mathbb{E}_x\left[f(x)^\top \mu_{y_x}\right] + \mathbb{E}_x\left[\log \sum_{k=1}^K \exp\left(f(x)^\top \mu_k\right)\right]$$

$$= - \mathbb{E}_x\left[f(x)^\top \mu_{y_x}\right] + \mathbb{E}_x\left[\log \frac{1}{K}\sum_{k=1}^K \exp\left(f(x)^\top \mu_k\right)\right] + \log K$$

$$= - \mathbb{E}_{(x,x^+)\in X^+}\left[f(x)^\top \mu_{y_x}\right] + \mathbb{E}_x\left[\log \mathbb{E}_{y^-}\left[\exp\left(f(x)^\top \mu_{y^-}\right)\right]\right] + \log K - \mathbb{E}_{(x,x^+)\in X^-}\left[f(x)^\top \mu_{y_x}\right]$$

$$\overset{(1)}{\geq} - \mathbb{E}_{(x,x^+)\in X^+}\left[f(x)^\top f(x^+) + f(x)^\top\left(\mu_{y_x} - f(x^+)\right)\right]$$

$$+ \mathbb{E}_x\left[\mathbb{E}_{y_x,\{y_i^-\}_{i=1}^M}\left[\log \frac{1}{M+1}\left(\exp\left(f(x)^\top \mu_{y_x}\right) + \sum_{i=1}^M \exp\left(f(x)^\top \mu_{y_i^-}\right)\right)\right]\right]$$

$$- \mathcal{O}\left(M^{-\frac{1}{2}}\right) + \log K - \mathbb{E}_{(x,x^+)\in X^-}\left[f(x)^\top f(x^+) + f(x)^\top\left(\mu_{y_x} - f(x^+)\right)\right]$$

$$\overset{(2)}{\geq} - \mathbb{E}_{(x,x^+)\in X^+}\left[f(x)^\top f(x^+)\right] - \mathbb{E}_{(x,x^+)\in X^+}\left[\|f(x) - \mu_{y_x}\|\right]$$

$$+ \mathbb{E}_x\left[\mathbb{E}_{y_x,\{y_i^-\}_{i=1}^M}\left[\log \frac{1}{M+1}\left(\exp\left(f(x)^\top \mu_{y_x}\right) + \sum_{i=1}^M \exp\left(f(x)^\top \mu_{y_i^-}\right)\right)\right]\right]$$

$$- \mathcal{O}\left(M^{-\frac{1}{2}}\right) + \log K - \mathbb{E}_{(x,x^+)\in X^-}\left[f(x)^\top f(x^+)\right] - \mathbb{E}_{(x,x^+)\in X^-}\left[\|f(x^+) - \mu_{y_x}\|\right]$$

$$= - \mathbb{E}_{x,x^+}\left[f(x)^\top f(x^+)\right] + \mathbb{E}_x\left[\mathbb{E}_{y_x,\{y_i^-\}_{i=1}^M}\left[\log \frac{1}{M+1}\left(\exp\left(f(x)^\top \mu_{y_x}\right) + \sum_{i=1}^M \exp\left(f(x)^\top \mu_{y_i^-}\right)\right)\right]\right]$$

$$- \mathbb{E}_{(x,x^+)\in X^+}\left[\|f(x) - \mu_{y_x}\|\right] - \mathbb{E}_{(x,x^+)\in X^-}\left[\|f(x^+) - \mu_{y_x}\|\right] - \mathcal{O}\left(M^{-\frac{1}{2}}\right) + \log K$$

$$= - \mathbb{E}_{x,x^+}\left[f(x)^\top f(x^+)\right] - \mathbb{E}_{(x,x^+)\in X^+}\left[\|f(x) - \mu_{y_x}\|\right] - \mathbb{E}_{(x,x^+)\in X^-}\left[\|f(x^+) - \mu_{y_x}\|\right] - \mathcal{O}\left(M^{-\frac{1}{2}}\right) + \log K$$

$$+ \mathbb{E}_x\left[\mathbb{E}_{y_x,\{y_i^-\}_{i=1}^M}\left[\log \frac{1}{M+1}\left(\mathbb{E}_{\left\{x^+,x_i^-\,|\,y_{x^+}=y_x,y_{x_i^-}=y_i^-\right\}}\left[\exp\left(f(x)^\top f(x^+)\right) + \sum_{i=1}^M \exp\left(f(x)^\top f(x_i^-)\right)\right]\right)\right]\right]$$

$$\overset{(3)}{\geq} - \mathbb{E}_{x,x^+}\left[f(x)^\top f(x^+)\right] - \sqrt{\mathbb{E}_{(x,x^+)\in X^+}\left[\|f(x) - \mu_{y_x}\|^2\right]} - \sqrt{\mathbb{E}_{(x,x^+)\in X^-}\left[\|f(x^+) - \mu_{y_x}\|^2\right]} - \mathcal{O}\left(M^{-\frac{1}{2}}\right) + \log K$$

$$+ \mathbb{E}_x\left[\mathbb{E}_{y_{\bar{x}},\{y_i^-\}_{i=1}^M}\left[\mathbb{E}_{\left\{x^+,x_i^-\,|\,y_{x^+}=y_x,y_{x_i^-}=y_i^-\right\}}\left[\log \frac{1}{M+1}\left(\exp\left(f(x)^\top f(x^+)\right) + \sum_{i=1}^M \exp\left(f(x)^\top f(x_i^-)\right)\right)\right]\right]\right]$$

$$- \frac{1}{2}V(z\,|\,z\in\{f(x^+),f(x^-)\}, y_{x^+}=y_x)$$

$$= - \mathbb{E}_{x,x^+}\left[f(x)^\top f(x^+)\right] + \mathbb{E}_x\left[\mathbb{E}_{x^+,\{x_i^-\}_{i=1}^M}\left[\log\left(\exp\left(f(x)^\top f(x^+)\right) + \sum_{i=1}^M \exp\left(f(x)^\top f(x_i^-)\right)\right)\right]\right]$$

$$- \sqrt{V(f(x))} - \sqrt{V^-(f(x))} - \frac{1}{2}V(z\,|\,z\in\{f(x^+),f(x^-)\}, y_{x^+}=y_x) - \mathcal{O}\left(M^{-\frac{1}{2}}\right) - \log\left(\frac{M+1}{K}\right)$$

$$= \mathcal{L}_{InfoNCE}(f) - \sqrt{V(f(x))} - \sqrt{V^-(f(x))} - \frac{1}{2}V(z\,|\,z\in\{f(x^+),f(x^-)\}, y_{x^+}=y_x) - \mathcal{O}\left(M^{-\frac{1}{2}}\right) - \log\left(\frac{M+1}{K}\right),$$

where the inequalities (1), (2) are similar to the proof of the above upper bound, the inequality (3) follows Lemma B.3 [8] and

---

[8]Wang et al. (2022) proved the convex function logsumexp is $\frac{1}{2}$-smooth.

the Cauchy–Schwarz inequality. We can get the lower bound of $\mathcal{L}_{CE}(g_{f,\mu}) + \log\left(\frac{M+1}{K}\right) - \mathcal{L}_{InfoNCE}(f)$ as

$$\mathcal{L}_{CE}(g_{f,\mu}) + \log\left(\frac{M+1}{K}\right) - \mathcal{L}_{InfoNCE}(f)$$
$$\geq -\sqrt{V\left(f(x)\right)} - \sqrt{V^-\left(f(x)\right)} - \frac{1}{2}V(z|z \in \{f(x^+), f(x^-)\}, y_{x^+} = y_x) - \mathcal{O}\left(M^{-\frac{1}{2}}\right).$$

$\square$

By integrating the Theorem 4.5 of Arora et al. (2019), we can directly derive Corollary C.2 which measures the upper bound of the classification risk $\mathcal{L}_{CE}(g_{f^*,W})$.

**Corollary C.2.** *Under the condition of Theorem 4.2, the downstream classification risk $\mathcal{L}_{CE}(g_{f^*,W}) + \log\left(\frac{M}{K}\right)$ can be upper bounded by $\mathcal{L}_{InfoNCE}(f^*) + \sqrt{V^-\left(f^*(x)\right)} + \sqrt{V\left(f^*(x)\right)} + \mathcal{O}\left(M^{-\frac{1}{2}}\right)$.*

**Proof of Corollary C.2:** According to Lemma B.4, we can know that, for any $f^* \in \arg\min_{f \in \mathcal{F}_1} \hat{\mathcal{L}}_{InfoNCE}(f)$ and $g \in \mathcal{F}_2$, the downstream classification risk $\mathcal{L}_{CE}(g_{f^*,W})$ with the linear classifier $W$ can be bounded by the mean downstream classification risk $\mathcal{L}_{CE}(g_{f^*,\mu})$ with the mean classifier $\mu$. And, from Theorem 4.2, we can obtain the upper bound of the mean downstream classification risk $\mathcal{L}_{CE}(g_{f^*,\mu})$ as $\mathcal{L}_{InfoNCE}(f^*) + \sqrt{V^-\left(f^*(x)\right)} + \sqrt{V\left(f^*(x)\right)} + \mathcal{O}\left(M^{-\frac{1}{2}}\right) - \log\left(\frac{M}{K}\right)$. Therefore, the upper bound of Corollary C.2 is proved. $\square$

**Assumption C.3** (Assumption 4.5 (restated)). Let a sample and its corresponding sample after applying SVD be represented as the matrices $X$ and $\hat{X}_q \in \mathbb{R}^{m \times m'}$, respectively. Assume that there are $q^*$ singular values associated with semantic-related information. When $q \geq q^*$, under Assumption 3.1 and the augmentation collection $\mathcal{T}$, the true label of the augmented sample of $\hat{X}_q$ is inconsistent with the latent label of $X$ with the probability $\alpha_q \leq \alpha$. Conversely, when $q < q^*$, the corresponding probability satisfies $\alpha_q > \alpha_{q^*}$.

**Statement of Assumption 4.5:** In practice, each sample possesses a unique amount of semantic-related information content. While for convenience, we assume that each sample can be decomposed into $q^*$ singular values capturing semantic-related information. Under Lemma 4.4 and the default assumption that there is a positive correlation between the amount of information and the importance of information, we can demonstrate that the larger the singular value, the more semantic-related the information captured in the corresponding subspace. Table 1 show the trend of increase of downstream classification accuracy from the experiment discarding the singular values $s_1, s_2$ to the one discarding $s_{31}, s_{32}$, which empirically validates this demonstration. Based on the above analysis, we can derive that truncated SVD firstly discards semantically irrelevant information that leads to labeling error. When $q$ takes the value $q^*$, samples don't have any semantically irrelevant information. When $q < q^*$, labeling error will increase due to the loss of semantic-related information.

$\square$

**Theorem C.4** (Theorem 4.7 (restated)). *Given the conditions of Theorem 4.2 and Assumption 4.6, after taking the truncated SVD on $\bar{\mathcal{D}}$, the mean downstream classification risk $\mathcal{L}_{CE}(g_{f,\mu}) - \mathcal{L}_{InfoNCE}(f)$ with the encoder $f$ can be upper bounded by*

$$\epsilon_{q^*} + \epsilon_q + \mathcal{O}\left(M^{-\frac{1}{2}}\right) - \log\left(\frac{M}{K}\right)$$

*and lower bounded by*

$$-\epsilon_{q^*} - \epsilon_q - \frac{1}{2}V(f(x^-)) - \mathcal{O}\left(M^{-\frac{1}{2}}\right) - \log\left(\frac{M+1}{K}\right).$$

*When $q = q^*$, the two bounds are $\epsilon_{q^*} + \mathcal{O}\left(M^{-\frac{1}{2}}\right) - \log\left(\frac{M}{K}\right)$ and $-\epsilon_{q^*} - V(f(x^-)) - \mathcal{O}\left(M^{-\frac{1}{2}}\right) - \log\left(\frac{M+1}{K}\right)$.*

**Proof of Theorem 4.7:**

**Upper bound:** From the proof of Theorem 4.2, we can get that

$$
\begin{aligned}
&\mathcal{L}_{InfoNCE}(f) \\
&\geq -\mathbb{E}_{(x,x^+)\in X^+}\left[f(x)^\top \mu_{y_x} + \left\|f(x^+) - \mu_{y_x}\right\|\right] + \mathbb{E}_x\left[\log \mathbb{E}_{y^-}\left[\exp\left(f(x)^\top \mu_{y^-}\right)\right]\right] \\
&\quad - \mathcal{O}\left(M^{-\frac{1}{2}}\right) + \log M - \mathbb{E}_{(x,x^+)\in X^-}\left[f(x)^\top \mu_{y_x} + \left\|f(x^+) - \mu_{y_x}\right\|\right] \\
&\geq \mathcal{L}_{CE}(g_{f,\mu}) - \mathbb{E}_{(x,x^+)\in X^+}\left[\|f(x) - \mu_{y_x}\|\right] - \mathcal{O}\left(M^{-\frac{1}{2}}\right) + \log\left(\frac{M}{K}\right) - \mathbb{E}_{(x,x^+)\in X^-}\left[\|f(x^+) - \mu_{y_x}\|\right] \\
&= \mathcal{L}_{CE}(g_{f,\mu}) - \mathbb{E}_{(x,x^+)\in X^+}\left[\left\|\mathbb{E}_{\{x^+|y_{x^+}=y_x\}}[f(x) - f(x^+)]\right\|\right] - \mathcal{O}\left(M^{-\frac{1}{2}}\right) + \log\left(\frac{M}{K}\right) \\
&\quad - \mathbb{E}_{(x,x^+)\in X^-}\left[\left\|\mathbb{E}_x\left[f(x^+) - f(x)\right]\right\|\right] \\
&\overset{(1)}{\geq} \mathcal{L}_{CE}(g_{f,\mu}) - \mathbb{E}_{(x,x^+)\in X^+}\left[\|f(x) - f(x^+)\|\right] - \mathcal{O}\left(M^{-\frac{1}{2}}\right) + \log\left(\frac{M}{K}\right) - \mathbb{E}_{(x,x^+)\in X^-}\left[\|f(x^+) - f(x)\|\right] \\
&\overset{(2)}{\geq} \mathcal{L}_{CE}(g_{f,\mu}) - \epsilon_{q^*} - \epsilon_q - \mathcal{O}\left(M^{-\frac{1}{2}}\right) + \log\left(\frac{M}{K}\right),
\end{aligned}
$$

where the inequality (1) is derived from the Cauchy–Schwarz inequality and the inequality (2) is due to Assumption 4.6.

Then,

$$
\mathcal{L}_{CE}(g_{f,\mu}) + \log\left(\frac{M}{K}\right) - \mathcal{L}_{InfoNCE}(f) \leq \epsilon_{q^*} + \epsilon_q + \mathcal{O}\left(M^{-\frac{1}{2}}\right).
$$

**Lower bound:** Similarly, we can get that

$$
\begin{aligned}
&\mathcal{L}_{CE}(g_{f,\mu}) \\
&\geq \mathcal{L}_{InfoNCE}(f) - \mathbb{E}_{(x,x^+)\in X^+}\left[\|f(x) - \mu_{y_x}\|\right] - \mathbb{E}_{(x,x^+)\in X^-}\left[\|f(x^+) - \mu_{y_x}\|\right] \\
&\quad - \frac{1}{2}V(z|z \in \{f(x^+), f(x^-)\}, y_{x^+} = y_x) - \mathcal{O}\left(M^{-\frac{1}{2}}\right) - \log\left(\frac{M+1}{K}\right) \\
&\geq \mathcal{L}_{InfoNCE}(f) - \epsilon_{q^*}^2 - \epsilon_q^2 - \frac{1}{2}V(z|z \in \{f(x^+), f(x^-)\}, y_{x^+} = y_x) - \mathcal{O}\left(M^{-\frac{1}{2}}\right) - \log\left(\frac{M+1}{K}\right).
\end{aligned}
$$

$\square$

**Corollary C.5.** *Under the condition of Theorem 4.7, the downstream classification risk $\mathcal{L}_{CE}(g_{f^*,W}) + \log\left(\frac{M}{K}\right)$ can be upper bounded by $\mathcal{L}_{InfoNCE}(f^*) + \epsilon_{q^*} + \epsilon_q + \mathcal{O}\left(M^{-\frac{1}{2}}\right)$. When $q = q^*$, the bound is $\mathcal{L}_{InfoNCE}(f^*) + \epsilon_{q^*} + \mathcal{O}\left(M^{-\frac{1}{2}}\right)$.*

**Theorem C.6** (Theorem 4.9 (restated)). *Let Assumption 3.1 hold. For the empirical optimal encoder $f^*$ taking the truncated SVD with hyper-parameter $q$ on unlabeled original dataset, there exists a linear head $W^* \in \mathbb{R}^{k \times K}$ with norm $\|W^*\|_F \leq 1/(1 - \lambda_{k,q})$ such that*

$$
\mathcal{E}(f^*, W^*) \leq \frac{4\alpha_q}{\lambda_{k+1,q}} + 8\alpha_q,
$$

*where $k$ denotes the dimension of embedding space and $\lambda_{k+1,q}$ denotes the $k+1$-th eigenvalues of $L_q$.*

**Proof of Theorem 4.9**: Lemma B.5 states that

$$
\Pr_{\bar{x}\sim\mathcal{P},x\sim p(\cdot|\bar{x})}[g_{f^*,W^*}(x) \neq y_{\bar{x}}] \leq \frac{4\alpha}{\lambda_{k+1}} + 8\alpha.
$$

We rewrite the definition $\mathcal{E}(f, W) = \Pr_{\bar{x}\sim\mathcal{P}}[g_{f,W}(\bar{x}) \neq y_{\bar{x}}]$ as $\Pr_{\bar{x}\sim\mathcal{P},x\sim p(\bar{x}|\bar{x})}[g_{f,W}(x) \neq y_{\bar{x}}]$. Thus, after taking the truncated SVD with hyper-parameter $q \in [m]$ on unlabeled sample, the following bound holds

$$
\mathcal{E}(f^*, W^*) = \Pr_{\bar{x}\sim\mathcal{P},x\sim p(\bar{x}|\bar{x})}[g_{f^*,W^*}(x) \neq y_{\bar{x}}] \leq \Pr_{\bar{x}\sim\mathcal{P},x\sim p(\cdot|\bar{x})}[g_{f^*,W^*}(x) \neq y_{\bar{x}}] \leq \frac{4\alpha_q}{\lambda_{k+1,q}} + 8\alpha_q.
$$

$\square$

## D. Experimental setting

**Details of models and datasets.** For the backbone structure, we use three variants of Resnet, i.e., Resnet-18, Resnet-50, and Resnet-152, where the dimensions of embedding are chosen from 128, 256, 512, 1024, and 2048. We use two-layers multilayer perceptron (MLP) to be the projection layers. For the datasets, we employ three benchmark datasets, i.e., CIFAR-10, CIFAR-100, and STL-10.

**Details of pre-training and fine-tuning.** We do pre-training for 100 epochs with batch size 256 using the optimizer Adam. The optimizer has the weight decay parameter 0.0004 and learning rate 0.0003. The learning rate decreases following the cosine schedule. In this stage, the dimension of the final output of the model is set as 128. For the downstream fine-tuning process, we train with a small number of labeling samples for 100 epochs with batch size 256 using the optimizer Adam with weight decay parameter 0.0008 and learning rate 0.0003. We use 1 RTX 2070 GPU for all experiments.

**Details of these experiments with inflation** Considering fair comparison, we do pre-training for 33 epochs with batch size 256 when we make data inflation. The settings of fine-tuning are the same for all experiments.

**Details of augmentations** In the main text, we provide several groups of augmentation strategies whose components are listed. We need to make the explanations that these abbreviations of augmentations without specific parameter settings adopt their default settings. For example, "RRC" (instead of "RRC(0.08, 0.5)") adopts the default parameters "(0.08, 1.0)", where the range "(0.08, 1.0)" denotes the proportion of preserved area after cropping. "Color jitter" (instead of "Color jitter(0.5, 0.4)") adopts the default parameters "(1.0, 0.8)", where "1.0" denotes a scale parameter and "0.8" denotes a probability parameter. "Cutout" (instead of "Cutout(0.5, 1.0)") adopts the default parameters "(0.1, 1.0)", where the range "(0.1, 1.0)" denotes the coordinate parameters used to cut samples.

## E. Discussions with related works

Huang et al. (2023b) proposed the concept of augmented distance and provided some upper bounds revealing the theoretical effect of augmented distance on understream classification performance. Specifically, they found that the classification performance of contrastive SSL is related to three key factors: alignment of positive samples, divergence of class centers, and concentration of augmented data. Theorem 2 in our work provided both upper and lower bound, which can not only give similar conclusions but also reveal some additional factors. Firstly, the term $V(f(x)|y_{\bar{x}})$ implies the alignment of positive samples. Secondly, the term $V_{y_{\bar{x}}^-}(f(x)|y_{\bar{x}})$ stems from labeling error caused by data augmentation, which is similar to the concentration of augmented data. Thirdly, the term $V(f(x^-)|y^-)$ implies the alignment of negative samples, which is not considered by Huang et al. (2023b). More importantly, we further improved the bounds of Theorem 2 via data dimensionality reduction and provided the corresponding theoretical analysis and empirical observations (Section 4.2).

Cui et al. (2023) established a theoretical framework for weakly supervised contrastive learning for the first time. Their results revealed that 1) semi-supervised information improves the error bound compared with purely unsupervised contrastive learning by using all labeled samples; 2) joint training of supervised and unsupervised contrastive learning does not improve the error bound compared with purely supervised or purely unsupervised contrastive learning. Although weakly supervised contrastive learning is not the topic of our work, the labeling error considered by our work is analogous to a type of weak supervision, i.e., noisy-labeled information. Therefore, we will extend the theoretical analysis of this work to weakly supervised contrastive learning in our future work. Besides, Cui et al. (2023) and our work both gave the suggestion that we should choose a moderate feature dimension $k$, which enhances the credibility of our suggestion.

## F. Limitations

The limitations of our work are listed as follows:

- This paper selects the embedding dimension $k = 512$ or $k = 1024$ as an example of a moderate choice. Nevertheless, we've observed that the optimal values of $k$ vary somewhat across different experimental settings, showing a degree of inconsistency. Delving into the underlying reasons for this inconsistency holds great interest since it has the potential to provide valuable insights that would be beneficial for the design of more effective algorithms.

- This paper pioneers in theoretically elucidating the role of labeling error in contrastive learning from a novel perspective, data dimensionality reduction. Algorithmically, we utilize the classical SVD for preliminary empirical validation. However, considering the unavailability and diversity of $p^*$ across different settings, the fixed $q$ in the traditional SVD

fails to guarantee a sufficiently small labeling error. Thus, our future research will focus on developing a more flexible low-rank image approximation method, which assigns a distinct truncated parameter $q$ to each sample.

- Even though our work mainly concentrates on the study of the potential false positive augmented samples, conducting research on false negative augmented samples remains of great necessity.

## G. Other Experimental Results

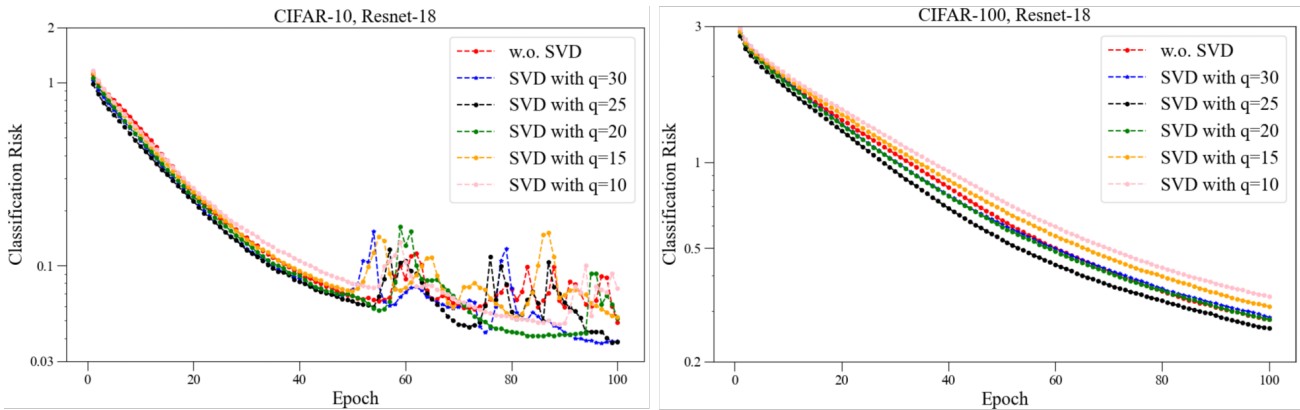

*Figure 5.* Classification risks of SimCLR using the augmentation $\mathcal{T}_1$ and the truncated SVD with different values of $q$. The corresponding top-1 accuracies are shown in Table 2.

*Table 8.* Downstream classification top-1 accuracies (%) of SimCLR ($\mathcal{L}_{InfoNCE}$) using the truncated SVD ($q = 30$ for CIFAR-10 and CIFAR-100, $q = 90$ for STL-10) with different embedding dimension $k$.

| $\mathcal{T}$ | Encoder | Dataset | Embedding Dimension | | | | |
| --- | --- | --- | --- | --- | --- | --- | --- |
| | | | $k = 128$ | $k = 256$ | $k = 512$ | $k = 1024$ | $k = 2048$ |
| $\mathcal{T}_1$ | Resnet-18 | CIFAR-10 | 68.12 | 69.11 | **69.48** | 69.27 | 68.84 |
| $\mathcal{T}_1$ | Resnet-50 | CIFAR-10 | 67.72 | 68.29 | **69.12** | 67.83 | 63.36 |
| $\mathcal{T}_1$ | Resnet-152 | CIFAR-10 | 66.75 | 67.64 | **68.05** | 65.98 | 62.50 |
| $\mathcal{T}_1$ | Resnet-18 | CIFAR-100 | 37.68 | 38.49 | **39.59** | 39.42 | 39.19 |
| $\mathcal{T}_1$ | Resnet-50 | CIFAR-100 | 38.96 | 39.50 | **39.83** | 38.10 | 32.23 |
| $\mathcal{T}_1$ | Resnet-18 | STL-10 | 71.28 | 71.37 | **71.93** | 71.49 | 71.34 |
| $\mathcal{T}_1$ | Resnet-50 | STL-10 | 72.72 | **73.76** | 72.53 | 71.73 | 70.28 |

*Table 9.* Downstream classification top-1 accuracies (%) of SimCLR ($\mathcal{L}_{InfoNCE}$) using the truncated SVD ($q = 30$ for CIFAR-10) with different epochs.

| $\mathcal{T}$ | Encoder | Dataset | SVD | Epochs | | | | |
|---|---|---|---|---|---|---|---|---|
| | | | | 100 | 200 | 300 | 400 | 500 |
| $\mathcal{T}_1$ | Resnet-18 | CIFAR-10 | w.o. SVD | 68.82 | 70.91 | 71.05 | 72.97 | 74.54 |
| $\mathcal{T}_1$ | Resnet-18 | CIFAR-10 | $q = 30$ | **69.48** | **71.06** | **71.48** | **73.43** | **74.94** |

*Table 10.* Downstream classification top-1 accuracies (%) of MoCo ($\mathcal{L}_{InfoNCE}$) using the truncated SVD with different $q$.

| $\mathcal{T}$ | Encoder | Dataset | w/o SVD | $q = 30$ | $q = 25$ | $q = 20$ | $q = 15$ | $q = 10$ |
|---|---|---|---|---|---|---|---|---|
| $\mathcal{T}_1$ | Resnet-18 | CIFAR-10 | 72.69 | **73.02** | 72.48 | 71.58 | 71.30 | 69.87 |

*Table 11.* Downstream classification top-1 accuracies (%) of SimCLR ($\mathcal{L}_{InfoNCE}$) using the truncated SVD on TinyImageNet-200 (image size $64 \times 64$, 50 pre-training epochs) and different backbones (ViT, ConvNeXt, 10 pre-training epochs) with different truncated parameter $q$.

| $\mathcal{T}$ | Encoder | Dataset | w/o SVD | $q = 60$ | $q = 50$ | $q = 40$ |
|---|---|---|---|---|---|---|
| $\mathcal{T}_1$ | Resnet-18 | TinyImageNet-200 | 28.72 | **29.38** | 28.44 | 27.94 |
| $\mathcal{T}_1$ | ViT | CIFAR-10 | 42.22 | **42.98** | 42.87 | 38.44 |
| $\mathcal{T}_1$ | ConvNeXt | CIFAR-10 | 55.75 | **56.37** | 55.87 | 55.59 |

*Table 12.* Downstream classification top-1 accuracies (%) of SimCLR ($\mathcal{L}_{InfoNCE}$) on CIFAR-10 using the truncated SVD with Random Erasing, GridMask and HidePatch.

| SVD | Encoder | Random Erasing | GridMask | HidePatch |
|---|---|---|---|---|
| w.o. SVD | Resnet-18 | 49.27 | 50.33 | 23.05 |
| $q = 30$ | Resnet-18 | **49.65** | **51.20** | **26.22** |

*Table 13.* Downstream classification top-1 accuracies (%) of SimCLR ($\mathcal{L}_{InfoNCE}$, 10 pre-training epochs) on ViT and ConvNeXt using the truncated SVD ($q = 30$ for CIFAR-10) with different embedding dimension $k$ ($-$ represents that it does not converge).

| $\mathcal{T}$ | Encoder | Dataset | Embedding Dimension | | | | | | |
|---|---|---|---|---|---|---|---|---|---|
| | | | $k = 128$ | $k = 256$ | $k = 512$ | $k = 1024$ | $k = 2048$ | $k = 3072$ | $k = 4096$ |
| $\mathcal{T}_1$ | ViT | CIFAR-10 | 41.62 | 41.28 | 42.22 | **43.14** | 40.18 | $-$ | $-$ |
| $\mathcal{T}_1$ | ConvNeXt | CIFAR-10 | 53.48 | 55.58 | 55.75 | 55.89 | **56.71** | 55.84 | 55.13 |

*Table 14.* Downstream classification top-1 accuracies (%) of BYOL ($\mathcal{L}_{InfoNCE}$, 10 pre-training epochs) using the truncated SVD ($q = 30$ for CIFAR-10) with different embedding dimension $k$.

| $\mathcal{T}$ | Encoder | Dataset | Embedding Dimension | | | | | | |
|---|---|---|---|---|---|---|---|---|---|
| | | | $k = 128$ | $k = 256$ | $k = 512$ | $k = 1024$ | $k = 2048$ | $k = 3072$ | $k = 4096$ |
| $\mathcal{T}_1$ | BYOL | CIFAR-10 | 33.71 | 33.85 | 34.08 | 34.09 | **34.32** | 33.11 | 33.00 |

*Table 15.* Downstream classification top-1 accuracies (%) of SimCLR ($\mathcal{L}_{InfoNCE}$) on CIFAR-100 and STL-10 using the truncated SVD with different $q$ or the data inflation strategy under the weak data augmentation adopted by Wang et al. (2024) ($\mathcal{T}_8 = \{$RRC(0.2, 1.0), Color jitter(0.5, 0.4), Random horizontal flip, Random grayscale, Gaussian blur$\}$).

| Dataset | $\mathcal{T}$ | Encoder | Inflation | w/o SVD | $q = 90$ | $q = 70$ | $q = 50$ |
|---|---|---|---|---|---|---|---|
| STL-10 | $\mathcal{T}_8$ | Resnet-18 | 70.86 | 70.51 | **71.49** | 71.23 | 69.26 |

| Dataset | $\mathcal{T}$ | Encoder | Inflation | | Inflation + ($q = 90$) | + ($q = 70$) | + ($q = 50$) |
|---|---|---|---|---|---|---|---|
| STL-10 | $\mathcal{T}_8$ | Resnet-18 | 70.86 | | **71.75** | 71.21 | 69.85 |

| Dataset | $\mathcal{T}$ | Encoder | Inflation | w/o SVD | $q = 30$ | $q = 25$ | $q = 20$ |
|---|---|---|---|---|---|---|---|
| CIFAR-100 | $\mathcal{T}_8$ | Resnet-18 | 42.84 | 42.29 | **42.93** | 42.72 | 42.5 |

| Dataset | $\mathcal{T}$ | Encoder | Inflation | | Inflation + ($q = 30$) | + ($q = 25$) | + ($q = 20$) |
|---|---|---|---|---|---|---|---|
| CIFAR-100 | $\mathcal{T}_8$ | Resnet-18 | 42.84 | | 43.04 | **43.08** | 42.92 |

*Table 16.* The estimated frequencies (%) of labeling error for CIFAR-10 with multiply augmentation strategies.

| Augmentation | RRC | $\mathcal{T}_1$ | $\mathcal{T}_2$ | $\mathcal{T}_3$ |
|---|---|---|---|---|
| Frequency | 27.9 | 37.5 | 48.3 | 42.1 |

*Table 17.* The cost of SVD on different datasets.

| Dataset | CIFAR-10 | TinyImageNet-200 | STL-10 |
|---|---|---|---|
| Size | $32 \times 32$ | $64 \times 64$ | $96 \times 96$ |
| Truncated parameter $q$ | $q = 30$ | $q = 60$ | $q = 90$ |
| Time of per image / s | 0.001994 | 0.004985 | 0.007978 |
| Time of all images / s | 99.7186 | 498.5094 | 797.7724 |

