# OpenReview forum: "How does Labeling Error Impact Contrastive Learning? A Perspective from Data Dimensionality Reduction"
_ICML.cc/2025/Conference — ICML 2025 poster_

### Official Review · Reviewer_ABgp · 2025-03-11

**Overall Recommendation:** 3

**Summary:**

In this paper, the authors provide a detail theoretical analysis on the effect of data augmentation on the downstream classification performance in contrastive learning, where the intra-class and inter-class augmentation overlap are considered. Based on these results, the authors propose to apply SVD on the input images and verify experimentally and theoretically that this could reduce the negative effect from inter-class augmentation overlap. Meanwhile, the authors also show that adopting SVD would also decrease the connectivity of the augmentation graph and then hurt the downstream classification performance. As a remedy, the authors propose to use moderate embedding dimension, which may increase the connectivity of the augmentation graph and then counteract the negative effects of SVD. The experimental results on benchmark datasets support the theoretical findings.

***
**Update after Rebuttal**

Thanks the authors for their detail responses, which have adequately addressed my concerns on the theoretical results. I encourage the authors to include these modified results and the discussions about those related work in the final version.

**Claims And Evidence:**

The motivation of this work is clear and reasonable. Applying SVD on the input images could filter out the semantic irrelevant information (background) while maintaining the key semantic information (as shown in Figure 4), which reduces the probability of the event that two inter-class images have overlap parts after augmentation (Figure 1) and thus help improve the downstream classification performance. However, since the semantic irrelevant information is remove, it is harder to obtain the same augmented images from the vanilla images by different augmented approaches, indicating that the connectivity of the augmentation graph is reduced. The above claims and intuitions are supported by the experimental results.

**Essential References Not Discussed:**

Most of the key studies on the theoretical analysis for contrastive learning are cited in this paper. However, there is also some work that need to be cited and discuss. In [1], the authors introduce the concept of augmented distance to depict the semantic similarity of two augmented images, which could also used to analyze the labeling error. In [2], the authors also analyze how the noisy-label information affect the down
stream classification error.

[1] Towards the Generalization of Contrastive Self-Supervised Learning. Huang et al., ICLR 2023.

[2] Rethinking Weak Supervision in Helping Contrastive Representation Learning. Cui et al., ICML 2023.

**Experimental Designs Or Analyses:**

From my view, the experimental design is reasonable. For the analysis, the authors mainly focus on the downstream classification performance of the model in this work. Indeed, the domain generalization ability is also important. It would be better if the authors could conduct some experiments to analyze how SVD and the moderate embedding dimension technique affects the domain generalization ability.

**Methods And Evaluation Criteria:**

The proposed method are evaluated on benchmark image datasets, which are also adopted in previous contrastive learning studies. Most of the experimental settings also followed previous ones. Therefore, the experimental results are sound and convinced.

**Other Comments Or Suggestions:**

Typo: "SLT-10" in the caption of Figure 4 should be "STL-10".

**Other Strengths And Weaknesses:**

The strength of this work is providing detailed theoretic analysis on the effects of intra-class and inter-class augmentation overlap on the downstream classification performance, which goes beyond previous work. As for the weakness, although applying SVD could mitigate the inter-class augmentation overlap, it also reduce the connectivity of the augmentation graph. Therefore, it is unclear whether the combination of applying SVD and using moderate embedding dimension would bring positive or negative effects on the downstream classification performance.

**Questions For Authors:**

1. You only consider downstream classification performance in this work. The domain generalization performance or domain transfer ability is also another important criteria. To what extent applying SVD or using moderate embedding dimension would affect the domain generalization performance of the model?

**Relation To Broader Scientific Literature:**

The key contribution of this work is establishing fine-grained theoretical analysis on the effects of data augmentation on the downstream classification performance in contrastive learning, particularly the effects from inter-class augmentation overlap. This could enhance the understanding to the mechanism of contrastive learning. Also, this work could also provide new insights to the computer vision community on how to learn representation via proper data augmentation. The moderate embedding dimension technique may also be helpful for researchers that apply contrastive learning on other formations of data, such as graphs or texts.

**Theoretical Claims:**

I have carefully checked all the proofs provided in the appendix. The core idea generally follows that of (Wang et al., 2022), and there is no major issues. However, I still have the following concerns: (1) In line 748-752, a constant factor $2$ appears suddenly in terms $\mathbb{E} _{p(x,y^{\neg} _{\bar{x}})}  [f(x)^{\top}  \mu _{\bar{x}}]$ and $\sqrt{\mathbb{E} _{p(x,y^{\neg} _{\bar{x}})}  [\Vert f(x)^{\top}  \mu _{\bar{x}} \Vert^2] }$. It's not clear to me why this constant factor appears. It seems that maintaining the original factor $1$ does not effect the final bounds. Besides, inequality (4) holds only when $\mathbb{E} _{p(x,y^{\neg} _{\bar{x}})}  [f(x)^{\top}  \mu _{\bar{x}}] \geq 0$. The authors should provide more detailed explanation on this. (2) In line 902-904, the authors claim that inequality (3) could be obtained by $\Vert f(x) - \mu _{y _{\bar{x}}} \Vert^2 = \Vert f(x) \Vert^2 + \Vert \mu _{y _{\bar{x}}} \Vert^2 -2 f(x)^\top \mu _{y _{\bar{x}}} \leq \epsilon^2_q$ and $\Vert f(x) \Vert \leq 1$. However, I can only conclude that $-2 f(x)^\top \mu _{y _{\bar{x}}} \leq \epsilon^2_q - (\Vert f(x) \Vert^2 + \Vert \mu _{y _{\bar{x}}} \Vert^2)$ holds. Moreover, if $\Vert f(x) \Vert \geq 1$ and $\Vert \mu _{y _{\bar{x}}} \Vert \geq 1$ hold, I can also conclude that $-2 f(x)^\top \mu _{y _{\bar{x}}} \leq \epsilon^2_q - 2$. But, inequality (3) holds only when $-2 f(x)^\top \mu _{y _{\bar{x}}} \geq \epsilon^2_q - 2 $ holds. (3) In line 910-915, there is no appearance of the term $\frac{1}{2} \epsilon^2_q$, yet it appears in line 923-925. Could the authors clarify this?

---

> ### Author Rebuttal · Authors · 2025-03-30
>
> We are grateful to you for your valuable comments and constructive suggestions.
>
> **Q1:** A constant 2 appears suddenly. Inequality (4) holds only when inner product is greater than 0.
>
> **A1:** Thanks. The constant 2 is not necessary. We have deleted this constant factor. Therefore, inequality (4) holds without the requirement that the inner product term is greater than 0. We have also corrected the corresponding result in Theorem 4.2.
>
> ***
> **Q2:** Line 902-904: why does inequality (3) hold?
>
> **A2:** Thanks. We have corrected the inequality (3) in the anonymous link: https://imgse.com/i/pEsQ5Pe. We have also corrected the corresponding result in Theorem 4.7.
>
> ***
> **Q3:** Line 923-925: $1/2\epsilon_q^2$ appears suddenly.
>
> **A3:** Thanks. We have corrected the formula from line 918 to 925 in the anonymous link: https://imgse.com/i/pEsQ5Pe. We have also corrected the corresponding result in Theorem 4.7.
>
> ***
> **Q4:** Some experiments of domain generalization ability.
>
> **A4:** Thanks. In our opinion, this question is about the domain transfer ability from pre-training data to downstream classification data. However, our work assumed the distribution of downstream classification data is the same as the distribution of pre-training data. Therefore, we were unable to study domain transfer ability. We will attempt to mild this assumption to analyze the impact of SVD on domain transfer ability from empirical and theoretical perspectives.
>
> If our understanding is wrong, please do not hesitate to contact us. We will make some more clear explanations at once.
>
> ***
> **Q5:** Cite some works.
>
> **A5:** In our modified manuscript, we have made some discussions about these works you mentioned.
>
> **[1]** proposed the concept of augmented distance and provided some upper bounds revealing the theoretical effect of augmented distance on understream classification performance. Specifically, they found that the classification performance of contrastive SSL is related to three key factors: **alignment of positive samples, divergence of class centers, and concentration of augmented data**. Theorem 2 in our work provided both upper and lower bound, which can not only give similar conclusions but also reveal some additional factors. **Firstly**, the term $V(f(x)|y_{\bar{x}})$ implies the alignment of positive samples. **Secondly**, the term $V_{y_{\bar{x}}^{\neg}}(f(x)|y_{\bar{x}})$ stems from labeling error caused by data augmentation, which is similar to the concentration of augmented data. **Thirdly**, the term $V(f(x^-)|y^-)$ implies the alignment of negative samples, which is not considered by [1]. **More importantly**, we further improved the bounds of Theorem 2 via data dimensionality reduction and provided the corresponding theoretical analysis and empirical observations (Section 4.2).
>
> **[2]** established a theoretical framework for weakly supervised contrastive learning for the first time. Their results revealed that 1) semi-supervised information improves the error bound compared with purely unsupervised contrastive learning by using all labeled samples; 2) joint training of supervised and unsupervised contrastive learning does not improve the error bound compared with purely supervised or purely unsupervised contrastive learning. Although weakly supervised contrastive learning is not the topic of our work, the labeling error considered by our work is analogous to a type of weak supervision, i.e., noisy-labeled information. Therefore, we will extend the theoretical analysis of this work to weakly supervised contrastive learning in our future work. Besides, [1] and our work both gave the suggestion that we should choose a moderate feature dimension $k$, which enhances the credibility of our suggestion.
>
> [1]Huang et al., Towards the Generalization of Contrastive Self-Supervised Learning. ICLR 2023.
>
> [2]Cui et al., Rethinking Weak Supervision in Helping Contrastive Representation Learning. ICML 2023.
>
> ***
> **Q6:** Whether the combination of SVD and moderate embedding dimension would bring positive or negative effects.
>
> **A6:** Thanks. This question is the concern stated in the remark of Theorem 4.9. We use a moderate embedding dimension to mitigate the reduction of the connectivity for the augmentation graph. According to our empirical observations, if we use a moderate embedding dimension, the positive effect of SVD on the downstream classification performance is not offset by the reduction of the connectivity for the augmentation graph at least. We don’t know whether a moderate embedding dimension can completely eliminate this reduction, which is a limitation of our work. We aim to further explore the impact of the moderate embedding dimension in our future work, which was mentioned in the first point of Limitation (Appendix E).
>
> ***
> **Q7:** "SLT-10"to "STL-10".
>
> **A7:** Thanks. We have corrected this typo.
>
> ***
> **Q8:** Domain generalization performance of the model?
>
> **A8:** Please see **A4**.

---

### Official Review · Reviewer_nEce · 2025-03-13

**Overall Recommendation:** 3

**Summary:**

This paper investigated theoretically the impact of labeling error on the downstream classification performance of contrastive learning. The authors demonstrate—both theoretically and empirically—that employing a moderate embedding dimension, data inflation, weak augmentation, and SVD fosters greater graph connectivity and reduces labeling error, ultimately improving downstream classification accuracy.

**Claims And Evidence:**

The claims made in the submission are supported by clear and convincing evidence except for the claim in line 87 and line 88:
“They didn’t verify whether the labeling error caused by the weak augmentation is sufficiently small”.
It seems that there is still no evidence showing whether the labeling error is sufficiently small in this paper.

**Essential References Not Discussed:**

NA

**Experimental Designs Or Analyses:**

I have carefully examined the experimental designs and analyses in Table 4 and Table 5, which incorporate various augmentations and data inflation. These experiments are both appropriate and necessary to illustrate the importance of combining data inflation, SVD, and weak augmentation. However, they appear to focus exclusively on CIFAR-10, which may seem inconsistent given that CIFAR-100 and STL-10 were utilized in the earlier experiments.

**Methods And Evaluation Criteria:**

The proposed methods and evaluation criteria make sense for the problem.

**Other Comments Or Suggestions:**

There is an issue in the conclusion section (line 437):
"ultimately improving improve model performance."
It is recommended to revise it to "ultimately improving model performance."

**Other Strengths And Weaknesses:**

Strengths: This paper has the potential to inspire further exploration of labeling errors in contrastive learning across other domains, such as NLP and graph learning.

Weaknesses: The conclusions of this paper may not be universally applicable, as certain factors still involve heuristic choices, including the degree of data inflation, the extent of weak augmentation, the number of dimensions selected for data reduction, and the domain of the datasets used.

**Questions For Authors:**

--There is still no evidence showing whether the labeling error is sufficiently small in this paper. (Comparing the comment “They didn’t verify whether the labeling error caused by the weak augmentation is sufficiently small” in line 87 and line 88)

-- It appears that the experiments are conducted on only three datasets, all within the computer vision domain. Will the conclusions of this paper also hold for other contrastive learning domains, such as NLP and graph learning? Given the scope of the experimental results, they may not be sufficient to generalize across all contrastive learning methods.

**Relation To Broader Scientific Literature:**

1.	 A notable aspect of this paper is its use of SVD to specifically address label mismatch in self-supervised settings.
2.	 This paper extends prior efforts that combine generative models (e.g., DDPMs) with contrastive learning but emphasizes that simply generating more data does not solve the labeling error problem unless mislabeling is curbed.

**Theoretical Claims:**

I have examined the theoretical claims and identified the following issues:

1.	In Assumption 4.6, there seems to be no clear definition of $\epsilon(\alpha_{q*})$  and  $\epsilon(\alpha_{q})$

2.	In Theorem 4.9, there is no clear explanation for why the formula $\varepsilon(f^*,W^*)\leq \frac{4\alpha_q}{\lambda_{k+1,q}}+8\alpha_q$  might hold.

---

> ### Author Rebuttal · Authors · 2025-03-30
>
> We are grateful to you for your valuable comments and constructive suggestions.
>
> **Q1:** Still no evidence showing whether the labeling error is sufficiently small in this paper.
>
> **A1:** Thanks for your constructive comment. Our work didn’t guarantee that the label errors are sufficiently small, which was discussed in the second point of the Limitations section in Appendix E (line 984). The primary objective of this study is to demonstrate that the label error under the setting of prior work [1] is not negligible. Our approach further reduced labeling error, as validated in Tables 5 and 6. Achieving sufficiently low label error remains an ongoing research objective.
>
> [1]Wang, Y., et al., Do generated data always help contrastive learning? ICLR, 2024.
>
> ***
> **Q2:** No clear definition of $\epsilon$.
>
> **A2:** Thanks. We have supplemented their clear definitions as follows. $\epsilon(\alpha_{q})$ is defined as the maximum distance between $f(x),f(x^+)$ when taking the truncated parameter $q$, where $(x,x^+)$ is a false positive sample pair. And, $\epsilon(\alpha_{q^*})$ is defined as the minimum distance between $f(x),f(x^+)$ for any truncated parameter values. In summary, $[\epsilon(\alpha_{q^*}), \epsilon(\alpha_{q})]$ is the range of value of $||f(x)-f(x^+)||$ for $(x,x^+)\sim p(x,x^+, y_{\bar{x}}^\neg)$.
>
> ***
> **Q3:** Why does the formula hold in Theorem 4.9?
>
> **A3:** The proof of the result for Theorem 4.9 was provided at the end of Appendix C. This result is quite similar to Lemma B.5, with the key difference being the definition of $E(f,W)$. In the proof, we have rewritten the form of this definition as $\underset{\bar{x}\in\bar{D},x\in p(\bar{x}|\bar{x})}{\mathrm{Pr}} \left[g_{f^*,W^*}(x) \neq y_{\bar{x}}\right]$ to facilitate the comparison with Lemma B.5. Obviously, $x\in p(\bar{x}|\bar{x})$ is included in $x\in p(\cdot|\bar{x})$. According to Lemma B.5, the result of Theorem 4.9 holds.
>
> ***
> **Q4:** The improvements on Table 4 and Table 5 appear to focus exclusively on CIFAR-10, which may seem inconsistent given that CIFAR-100 and STL-10 were utilized in the earlier experiments.
>
> **A4:** Thanks. We have added related experiments on CIFAR-100 and STL-10 (anonymous link: https://imgse.com/i/pEsQI8H).
>
> ***
> **Q5:** The conclusions of this paper may not be universally applicable, as certain factors still involve heuristic choices, including the degree of data inflation, the extent of weak augmentation, the number of dimensions selected for data reduction, and the domain of the datasets used.
>
> **A5:** Thanks. We have made some explanations about the several factors you mentioned.
>
> **1) Data Inflation:** Data inflation is not the focus of this paper. Previous work [1] has shown that the more similar the distribution of inflated data is to the one of original data, the better the model performance. Therefore, this paper adopts the optimal inflation setting from [1].
>
> **2) Weak Augmentation:** We also adopt the weak augmentation strategy suggested in [1].
>
> **3) Data Dimensionality Reduction:** The SVD truncation parameter used in this work is manually set to verify the effectiveness of SVD, not to obtain optimal model performance. Therefore, we set a large $q$ value in most experiments. In the future, we will adaptively learn the optimal truncation value $q^*$, which is mentioned in the second point of Limitation in Appendix E.
>
> **4) Data Domain:** The experiments and theoretical analysis of this work are focused on the field of computer vision. Exploring whether the findings generalize to domains such as NLP and graph learning is an interesting and open question. Future work will extend this investigation to NLP and graph learning to assess the broader applicability of the proposed framework.
>
> [1]Wang, Y., et al., Do generated data always help contrastive learning? ICLR, 2024.
>
> ***
> **Q6:** "ultimately improving improve model performance" to "ultimately improving model performance"
>
> **A6:** Thanks. We have corrected this issue.
>
> ***
> **Q7:** Still no evidence showing whether the labeling error is sufficiently small in this paper.
>
> **A7:** Please see **A1**.
>
> ***
> **Q8:** Will the conclusions of this paper also hold for other contrastive learning domains, such as NLP and graph learning?
>
> **A8:** Thanks. The experiments and theoretical analysis of this work are focused on the field of computer vision. Exploring whether the findings generalize to domains such as NLP and graph learning is an interesting and open question. Future work will extend this investigation to NLP and graph learning to assess the broader applicability of the proposed framework.

---

> > ### Comment · Reviewer_nEce · 2025-04-07
> >
> > Thank the authors for their response. I have increased my score.

---

> > > ### Author Response · Authors · 2025-04-07
> > >
> > > Thank you very much for your recognition and support of our work

---

### Official Review · Reviewer_339q · 2025-03-14

**Overall Recommendation:** 3

**Summary:**

This paper theoretically analyzes the effect of labeling errors, particularly cases where an augmented example may belong to a different class than the original example. Based on this, the authors further study how performing dimensionality reduction on representations can mitigate the negative impact of labeling errors and support their findings with experiments.

**Claims And Evidence:**

The claims made in the submission are supported by clear and convincing evidence.

**Essential References Not Discussed:**

I didn’t notice any.

**Experimental Designs Or Analyses:**

The experimental design is valid and demonstrates the effect of applying SVD to the inputs.

**Methods And Evaluation Criteria:**

The evaluation makes sense for the problem.

**Other Comments Or Suggestions:**

I don’t have other comments.

**Other Strengths And Weaknesses:**

I think the paper makes a novel point by considering labeling errors caused by augmentation that might alter an example’s semantic meaning.

However, I find the practical contribution somewhat limited. First, demonstrating the negative impact of labeling errors seems quite intuitive—everyone would expect performance to worsen when this issue is present. So, the more practical contribution is likely the authors’ demonstration that applying SVD to inputs can mitigate the issue. However, it seems that there could be other straightforward solutions. For example, since the labeling issue ultimately stems from imperfections in the augmentation process, could it be addressed simply by making more careful choices in augmentation? For instance, setting the cropping ratio appropriately or relying on smarter augmentation techniques like AutoAugment.

Additionally, I feel there should be experiments to justify how frequently this type of labeling error occurs in real-world scenarios.

**Questions For Authors:**

My main questions are listed in the Strengths and Weaknesses section.

**Relation To Broader Scientific Literature:**

The paper is related to the literature on contrastive learning. It adds a new perspective by considering the possibility that data augmentation may transform an example into something that semantically belongs to another class.

**Theoretical Claims:**

I didn’t go through all the proofs in detail, but the theoretical results seem convincing to me.

---

> ### Author Rebuttal · Authors · 2025-03-30
>
> We are grateful to you for your valuable comments and constructive suggestions.
>
> **Q1:** Since the labeling issue ultimately stems from imperfections in the augmentation process, could it be addressed simply by making more careful choices in augmentation? For instance, setting the cropping ratio appropriately or relying on smarter augmentation techniques like AutoAugment.
>
> **A1:** Thanks for your constructive comment.
>
> **Firstly**, data augmentation presents a dual challenge in deep learning systems: excessive augmentation introduces label noise due to over-distorted samples, while insufficient augmentation compromises model performance through limited diversity. **Secondly**, the optimal augmentation method varies across different datasets. Optimal augmentation selection requires non-trivial domain-specific expertise and extensive empirical validation.
>
> Besides, some smarter augmentation techniques like AutoAugment can improve model performance by dynamically adjusting augmentation strategies. However, their time-consuming and resource-intensive nature limits practical implementation. In contrast, this work pioneers a novel paradigm by employing data dimensionality reduction as a pre-process strategy to improve model performance effectively. Empirical and theoretical results validate the effect of dimensionality reduction. We will conduct research on dimensionality reduction to develop a more flexible low-rank approximation method as a new augmentation method to achieve greater model performance, which was mentioned in the second point of Limitation (Appendix E).
>
> ***
> **Q2:** Additionally, I feel there should be experiments to justify how frequently this type of labeling error occurs in real-world scenarios.
>
> **A2:** Thanks. We have added experiments on multiply augmentation combinations to justify how frequently this type of labeling error occurs in real-world scenarios (anonymous link: https://imgse.com/i/pEsQo2d).

---

### Official Review · Reviewer_JnUb · 2025-03-14

**Overall Recommendation:** 3

**Summary:**

This paper investigates the impact of labelling errors introduced by augmentation in contrastive learning. The authors first demonstrate labelling errors in terms of positive and negative pairs and theoretically prove that these errors affect the upper and lower bounds of the downstream classification risk. Furthermore, they propose adopting Singular Value Decomposition (SVD) in the augmentation strategy to reduce irrelevant semantic features and minimise classification errors.

**Claims And Evidence:**

Please see comments regarding the experiment designs and weaknesses.

**Essential References Not Discussed:**

The paper discusses the significance of applying dimensionality reduction to contrastive learning embeddings to mitigate the impact of labelling errors. Beyond SVD, various feature extraction and dimensionality reduction techniques exist, which should have been discussed in the related works.

**Experimental Designs Or Analyses:**

- In Table 1, why do the authors present the performance of discontinuous singular value pairs (e.g., no evaluation of $s_{2,3}$, $s_{12,13}$)
- Is a 0.27% (row 4 column 4&5 in Table 2 results) statistically significant? In other words, are the improvements, which often appear to be less than 1%, when adopting SVD truly meaningful?
- For the ablation studies in the Appendix, why was $q=30$ used for all experiments when Table 2 shows that $q=25$ performs better on CIFAR-10?
- Have other augmentation combinations (besides RRC, Cutout, Color Jitter, and their combinations) been tested, and do they also support the claims made in this paper?

**Methods And Evaluation Criteria:**

I would appreciate experiments on ImageNet as well to demonstrate the generalisability of the proposed SVD-based augmentation.

**Other Comments Or Suggestions:**

Please see the weaknesses above.

**Other Strengths And Weaknesses:**

Strengths:

- The study examines the effects of varying the top q singular values, labelling error rates, model architectures, and embedding dimensions across three ResNet architectures and on three benchmarks.
- The authors provide theoretical proofs to support their claims.

---

Weaknesses:

- Some results require further justification (see comments on Experimental Designs).
- Certain experimental settings remain unclear, making the claims less convincing.
- There is no justification for the additional computational cost incurred by computing SVD as part of the proposed augmentation strategy.
- Not enough validation to demonstrate the generalisability of proposed method on different architectures and datasets.

**Questions For Authors:**

1. Is it possible to test your method on other augmentation combination (except RRC, Cutout, Colout jitter, and their combinations) to support the claims in this paper?
2. How to define *moderate* embedding dimensions on different architectures (for example the ViTs) and different contrastive learning methods (except SimCLR and MoCo)
3. Are the improvements of using truncated SVD considered significant?

I am open to revise my recommendation if the authors can provide more justification of the generalisability and significance of their method.

**Relation To Broader Scientific Literature:**

This paper contributes to the growing body of research on contrastive learning by addressing the realistic challenge of labelling errors, which has been largely overlooked in previous studies. The key contributions align with and extend prior findings in the following ways:

- The paper underscore the impact of labelling error in contrastive learning, providing a more realistic assumption compared to many existing works.

- The authors theoretically prove the negative impacts of labelling errors, which raise awareness for future research on to robustness and reliability of contrastive learning applications.

- This paper is closely related to existing works that investigate the effectiveness of adopting generated data and weak augmentation in contrastive learning. It extends these observation by demonstrating the benefit of using SVD is on par with adopting data inflation.

**Theoretical Claims:**

I reviewed the theoretical claims in Sections 3, 4.1, and 4.2, and they appear to be correct.

---

> ### Author Rebuttal · Authors · 2025-03-30
>
> We are grateful to you for your valuable comments and constructive suggestions.
>
> **Q1:** ImageNet?
>
> **A1:** Thanks. Considering the size of the original ImageNet is too large, it is hard to complete the experiment on it before the deadline (March 31 (AoE)). We have added experiments on TinyImageNet (anonymous link: https://imgse.com/i/pEsQWVK).
>
> ***
> **Q2:** Discontinuous singular value in Table 1?
>
> **A2:** We have added experiments for $s_{i,j}$ (link: https://imgse.com/i/pEsasuF).
>
> ***
> **Q3:** Are improvements meaningful?
>
> **A3:** Although a few experimental results did not demonstrate significant improvement, our all experiments exhibited a consistent trend of progress across different settings. In some cases, the improvement exceeded 1%, even when q=30 (such as columns 3,5,6 in Table 4). Further reductions in the q value may lead to additional performance gains.
>
> ***
> **Q4:** Why q=30 not q=25?
>
> **A4:** Since the optimal q varies across different experimental settings (Table 2), we fixed q=30 for subsequent experiments to ensure consistent improvement in downstream performance across varying conditions, thereby validating our theory, even though the improvement might be relatively limited in some settings when adopting q=30. Note that, this paper aims not to obtain optimal downstream performance but to investigate how dimensionality reduction affects labeling error.
>
> ***
> **Q5:** Other augmentations?
>
> **A5:** We have added experiments with Random Erasing, GridMask, and HidePatch (link: https://imgse.com/i/pEs15jA). They also support the claims made in this paper.
>
> ***
> **Q6:** Discussion of dimensionality reduction?
>
> **A6:** Thanks. We have supplemented the discussion of various dimensionality reduction techniques. Beyond SVD, there are various techniques for feature extraction and dimensionality reduction, including matrix and tensor decomposition (PCA[1], NMF[2]), dictionary learning[3], compressed sensing[4], deep learning (Autoencoder[5], GAN[6]). Since we use SVD as a simple example to study the effect of dimensionality reduction on contrastive learning, we didn’t carefully make a survey about various dimensionality reduction methods. Our empirical observations and theory show that an effective data augmentation based on dimensionality reduction is necessary. We will conduct research on dimensionality reduction to develop a flexible low-rank approximation as a new augmentation to achieve greater model performance, which was mentioned in the second point of Limitation (Appendix E).
>
> [1]Y. Ren, et al., Hyperspectral Image Spectral-Spatial Feature Extraction via Tensor Principal Component Analysis, Geoscience and Remote Sensing Letters, 2017.
>
> [2]M. Chen, et al., Feature Weighted Non-Negative Matrix Factorization, Transactions on Cybernetics, 2023.
>
> [3]P. Song et al., Multimodal Image Denoising Based on Coupled Dictionary Learning, International Conference on Image Processing (ICIP), 2018.
>
> [4]T. Hong, et al., A Complex Quasi-Newton Proximal Method for Image Reconstruction in Compressed Sensing MRI, Transactions on Computational Imaging, 2024.
>
> [5]Y. Shen, et al., DRACO: A Denoising-Reconstruction Autoencoder for Cryo-EM, NeurIPS, 2024.
>
> [6]D. Chen, et al., SSL: A Self-similarity Loss for Improving Generative Image Super-resolution, ACM MM, 2024.
>
> ***
> **Q7:** Experimental settings?
>
> **A7:** To ensure reproducibility, we have updated the experimental details in Appendix D of our revised manuscript.
>
> **Hardware:** All experiments are executed on an RTX 2070 GPU with an Intel(R) i7-10750H CPU.
>
> **Software:** Python 3.7.4 with PyTorch 1.13.1.
>
> **Optimizer, epoch, batchsize:** We provided them in the second point of Appendix D. Note that we pre-train STL-10 with 50 epochs due to its substantial data volume.
>
> **Augmentation:** We provided the parameter of every augmentation in the fourth point of Appendix D.
>
> **Supplementary experiments in our response:** Complete implementation details are included in their respective response.
>
> Should any additional technical specifications be required? Please inform us and we will promptly provide the requested information.
>
> ***
> **Q8:** Cost from SVD.
>
> **A8:** We have added the cost of SVD on different datasets (link: https://imgse.com/i/pEsQfUO).
>
> ***
> **Q9:** Validation on different backbones and datasets.
>
> **A9:** We have added experiments on TinyImageNet and ViT, ConvNeXt (link: https://imgse.com/i/pEsQWVK).
>
> ***
> **Q10:** Other augmentation?
>
> **A10:** Please see **A5**.
>
> ***
> **Q11:** Define moderate embedding dimensions.
>
> **A11:** We have added experiments on ViT, ConvNeXt, and BYOL (link: https://imgse.com/i/pEswcf1). We find that the optimal dimension is usually in $[512, 2048]$. The impact of the moderate dimension is valuable to be further explored in our future work, which was mentioned in the first point of Limitation (Appendix E).
>
> ***
> **Q12:** Are the improvements significant?
>
> **A12:** Please see **A3**.

---

> > ### Comment · Reviewer_JnUb · 2025-04-06
> >
> > I would like to thank the authors for clarifying my concerns, particularly regarding Q1, Q5, and Q9. I have reviewed the discussion between the authors and the other reviewers, and I share a concern raised by Reviewer nEce: the paper does not provide a broadly applicable guideline for generalising the proposed method. I am not fully convinced by the authors’ response that the parameters of SVD is mainly to *“verify the effectiveness of SVD, not to obtain optimal model performance”*, which emphasises the paper’s theoretical focus and defers the investigation of more general designs to future work. Nonetheless, I believe this limitation does not significantly detract from the paper’s contribution, given its theoretical foundation and experimental results as support. Therefore, I am inclined to improve my recommendation.

---

> > > ### Author Response · Authors · 2025-04-07
> > >
> > > Thank you very much for your recognition and support of our work

---

### Decision · Program_Chairs · 2025-05-01

**Decision:**

Accept (poster)

**Comment:**

This paper discusses the labeling error issues in contrastive learning and proposes a remedy through SVD on the inputs. While some reviewers initially had concerns about the scalability and applicability of the proposed method, the authors provided effective feedback with larger scale experiments on TinyImageNet, enhancing the practical applicability. Thus I recommend this paper for acceptance. Since the reviewers provided important feedback on the paper, I expect the authors to incorporate them into the final version.